# Single-cell transcriptomics reveals functionally specialized vascular endothelium in brain

Hyun-Woo Jeong[1]*, Rodrigo Diéguez-Hurtado[1], Hendrik Arf[1], Jian Song[2,3], Hongryeol Park[1], Kai Kruse[1], Lydia Sorokin[2,3], Ralf H Adams[1,3]*

[1]Department of Tissue Morphogenesis, Max Planck Institute for Molecular Biomedicine, Münster, Germany; [2]Institute of Physiological Chemistry and Pathobiochemistry, University of Münster, Münster, Germany; [3]Cells-in-Motion Interfaculty Centre (CiMIC), University of Münster, Münster, Germany

**Abstract** The blood-brain barrier (BBB) limits the entry of leukocytes and potentially harmful substances from the circulation into the central nervous system (CNS). While BBB defects are a hallmark of many neurological disorders, the cellular heterogeneity at the neurovascular interface, and the mechanisms governing neuroinflammation are not fully understood.

Through single-cell RNA sequencing of non-neuronal cell populations of the murine cerebral cortex during development, adulthood, ageing, and neuroinflammation, we identify reactive endothelial venules, a compartment of specialized postcapillary endothelial cells that are characterized by consistent expression of cell adhesion molecules, preferential leukocyte transmigration, association with perivascular macrophage populations, and endothelial activation initiating CNS immune responses. Our results provide novel insights into the heterogeneity of the cerebral vasculature and a useful resource for the molecular alterations associated with neuroinflammation and ageing.

**\*For correspondence:**
hyun-woo.jeong@mpi-muenster.mpg.de (H-WooJ);
ralf.adams@mpi-muenster.mpg.de (RHA)

**Competing interest:** The authors declare that no competing interests exist.

## Editor's evaluation

This study provides insight into an understudied cell type in the neurovascular unit involved in inflammatory disease and provides a resource of scRNA seq data for non-neuronal CNS cells. Further, it provides some interesting areas for future investigations into neuroimmune mechanisms used by the brain vasculature to control leukocyte transmigration in various health and disease conditions. This work will be of interest to vascular and neurovascular biologists, aging biologists, immunologists, and translational/clinical scientists interested in disease therapies.

## Introduction

The blood-brain barrier (BBB) controls the entry of a wide range of molecules from the circulation into the central nervous system (CNS) and thereby maintains the appropriate chemical and cellular composition of the neuronal 'milieu', which is required for the correct function of synapses and neuronal circuits (*Zlokovic, 2008*; *Zlokovic, 2010*). The BBB also protects the brain against the entry of leukocytes and potentially harmful substances, a function that is compromised in conditions such as multiple sclerosis, cancer, after stroke, or in response to physical brain damage (*Arvanitis et al., 2020*; *Lopes Pinheiro et al., 2016*; *Thal and Neuhaus, 2014*). The neurovascular unit, the anatomical structure underlying the BBB, comprised different cell types including endothelial cells (ECs), pericytes, and astrocytes, which are all located in close proximity and presumably affect each other through reciprocal interactions (*Armulik et al., 2011*; *Liebner et al., 2018*).

Previous work has established that leukocyte extravasation – the exit from the blood stream into the tissue – generally occurs in postcapillary venules of the skin, muscle, and mesentery, whereas in lung and liver this process is confined to microcapillaries (*Strell and Entschladen, 2008*). Even in the absence of neuroinflammation, peripherally activated circulating T cells can cross the endothelium of postcapillary vessels and reach the adjacent subarachnoid space. Here, T cells can encounter tissue resident antigen-presenting cells (APCs), but they are unable to traverse the astrocytic basement membrane and glia limitans (*Mapunda et al., 2022*). In the absence of cognate antigen presented by APCs, T cells undergo apoptosis or re-enter the circulation (*Mastorakos and McGavern, 2019*). Accordingly, immune surveillance and the initiation of CNS immune responses are highly dependent on the cellular and molecular components of the postcapillary venules and the perivascular environment, but we currently lack a comprehensive understanding of the underlying molecular mechanisms.

Transcriptional profiling has recently provided important insights into the cellular composition of the CNS and the arterial-venous zonation of brain vessels during postnatal development and adult homeostasis (*Sabbagh et al., 2018*; *Vanlandewijck et al., 2018*; *Zeisel et al., 2015*). On the other hand, brain ECs demonstrate similar gene expression changes in various BBB dysfunction models, suggesting a common mechanism for compromised BBB function throughout different neurological disorders (*Munji et al., 2019*). Despite these important insights, the molecular heterogeneity of vascular cells and their dynamic modulation, which is critical for BBB function (*Villabona-Rueda et al., 2019*), remains insufficiently understood. One critical technical issue is the low abundance of vascular cells relative to neural cell types in the brain.

In this study, we have established a protocol for the depletion of neurons and oligodendrocytes prior to droplet-based single-cell transcriptome analysis of non-neuronal cells in mouse cerebral cortex. We have generated a comprehensive resource of vascular gene expression at single-cell resolution during postnatal development, adulthood, ageing, and in the demyelinating neuroinflammatory condition of experimental autoimmune encephalomyelitis (EAE), a mouse model of human multiple sclerosis (*Ben-Nun et al., 2014*). This resulting data, which can be interrogated at single-cell.mpi-muenster.mpg.de, provides useful insights into the cellular and molecular heterogeneity of blood vessels in brain and permits identification of functionally specialized reactive postcapillary venules (REVs), which we propose to regulate the infiltration of activated leukocytes during neuroinflammation. Using a computational framework, we have also constructed a detailed cell-to-cell intercommunication map of the brain vasculature.

## Results

### Single-cell RNA-sequencing of non-neuronal cell population in mouse cerebral cortex

To enrich vascular and vessel wall-associated cell types while preserving the heterogeneity of these populations, we depleted myelin-associated neurons and oligodendrocytes, the most abundant cell types in CNS, using a bovine serum albumin (BSA)-gradient method (*Figure 1—figure supplement 1A, B*). Following quality control and data trimming (*Figure 1—figure supplement 1C*), a total of 29,406 single cells from three different postnatal ages, juvenile (10,796 cells, postnatal day 10), adult (9871 cells, 7–11 weeks), and aged (8739 cells, 18 months) were analyzed further (*Figure 1—figure supplement 2A*). Analysis by a nonlinear dimensionality reduction technique, uniform manifold approximation and projection (UMAP) (*Figure 1A–C*), and unsupervised hierarchical clustering (*Figure 1D and E*) established six different non-neuronal cell types, namely ECs, microglia (Micro), astroependymal cells (Astro), perivascular macrophages (PVMs), mural cells (Mural), and cerebral fibroblast-like cells (Fibro). Each cell type was successfully annotated by known marker genes such as *Flt1* for ECs, *Tmem119* for microglia, *Folr1* for astroependymal cells, *Mrc1* for PVMs, *Rgs5* for mural cells, and *Nov* for the Fibro population (*Figure 1E and F*). All these markers are expressed specifically and continuously in all age groups analyzed (*Figure 1—figure supplement 2B-D*).

### Unsupervised clustering reveals a diversity of EC subtypes

To gain insight into the heterogeneity of vascular and perivascular cell types, we first analyzed ECs during postnatal development using UMAP dimensionality reduction. Juvenile ECs were annotated to six different subclusters corresponding to arterial, capillary/precapillary arteriolar (CapA), capillary/

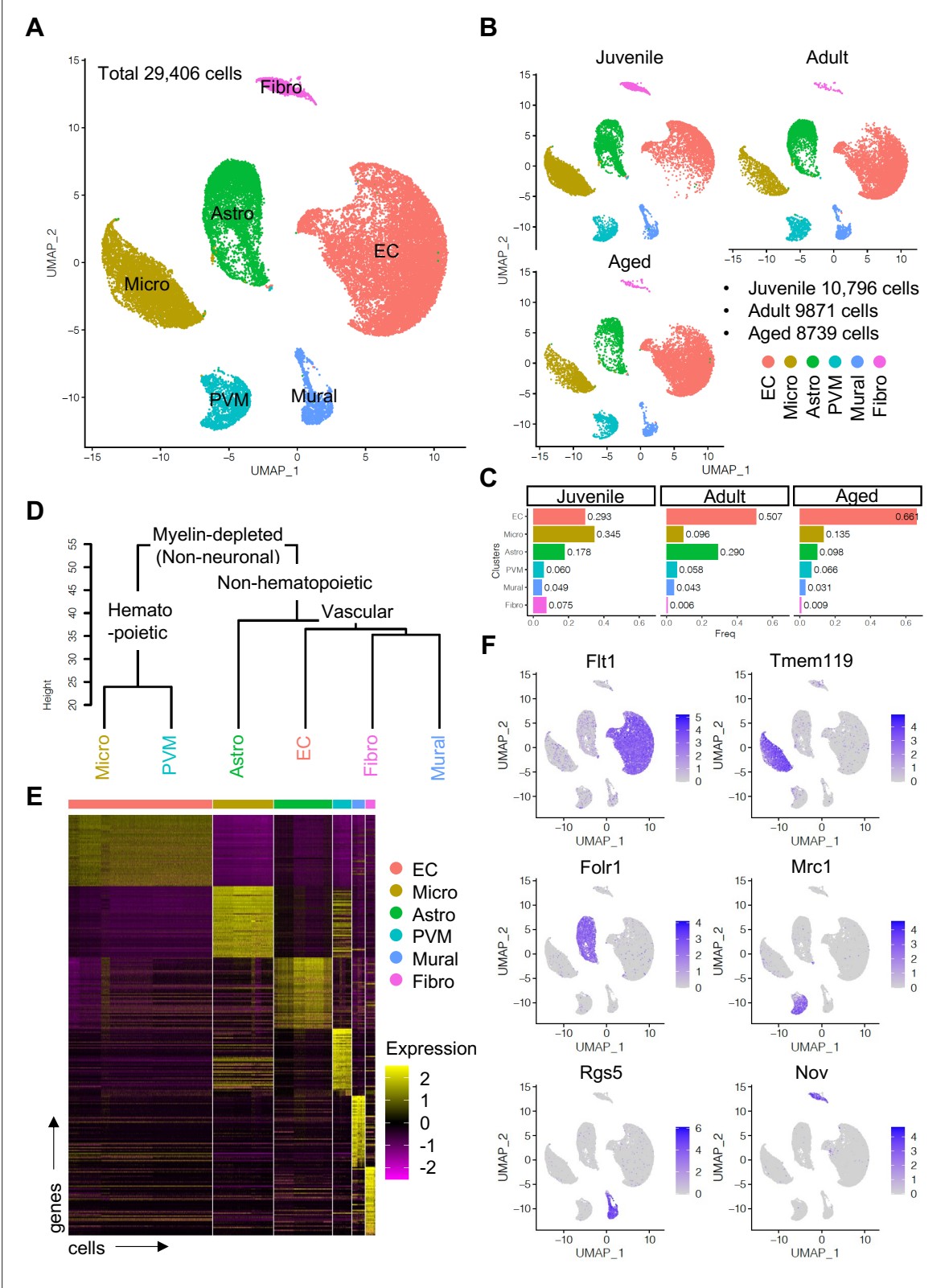

**Figure 1.** Single-cell RNA-seq analysis of non-neuronal cell types in mouse cerebral cortex. (**A**) Uniform manifold approximation and projection (UMAP) plot of 29,406 myelin-depleted single cells from murine cerebral cortex. Colors represent endothelial cells (EC), microglia (Micro), astroependymal cells (Astro), perivascular macrophage (PVM), mural cell (Mural), and cerebral fibroblast (Fibro). (**B**) Split UMAP plots showing separated cells from juvenile, adult, and aged samples, respectively. (**C**) Bar plots show the relative frequency of cell types for each age. (**D**) Dendrogram describing the taxonomy

*Figure 1 continued on next page*

Figure 1 continued

of all identified non-neuronal cell types. (**E**) Heatmap indicates the top 50 marker genes for each cell type. (**F**) Expression distribution of the top marker genes for each cell type projected onto the UMAP plot. Color represents the scaled expression level.

The online version of this article includes the following source data and figure supplement(s) for figure 1:

**Source data 1.** Source data for *Figure 1C*.

**Figure supplement 1.** Analysis of vascular and perivascular cell types in mouse cerebral cortex using single-cell RNA-seq.

**Figure supplement 2.** Expression of major cell type marker genes.

postcapillary venular (CapV), venous, mitotic, and tip ECs (*Figure 2—figure supplement 1A, B*). Trajectory analysis indicates that there are one mitotic axis (M) and a tip cell axis (T), which are closely associated with venous populations (V), whereas arterial ECs (A) are found in a different branch of polarization (*Figure 2—figure supplement 1C*). Distinct transcriptional profiles reveal the expression of venous genes in tip and mitotic ECs (*Figure 2—figure supplement 1D, E*), while these two populations are also distinguished by specifically expressed transcripts such as *Mcam* for tip cells and *Top2a* for mitotic cells (*Figure 2—figure supplement 1F*). Pseudotime reconstruction analysis reveals a continuum of venous-to-arterial gene expression and also confirmed the presence of venous attributes in tip and mitotic ECs (*Figure 2—figure supplement 1G-J*), consistent with previous findings that tip and mitotic ECs emerge from veins and are, in part, incorporated into arteries (*Pitulescu et al., 2017*; *Xu et al., 2014*).

Next, we analyzed adult and aged EC populations. Although there was no striking change in gene expression profiles between adult and aged ECs (Spearman's correlation coefficient = 0.9852), 407 genes were identified as differentially expressed (Padj <0.001, *Figure 2—figure supplement 2A*). Gene set enrichment analysis (GSEA) indicates that genes related to ECM-receptor interaction, focal adhesion, and the P53 signalling pathway are enriched in adult ECs, while genes related to cell adhesion molecules, chemokine signalling, and T cell receptor signalling are enriched in aged ECs (*Figure 2—figure supplement 2B, C*), indicating that ECs undergo molecular changes toward a proinflammatory state during ageing, which is consistent with recently published work (*Chen et al., 2020*).

To investigate EC subtypes during homeostasis, we performed subclustering analysis of adult and aged ECs together. As expected, tip and mitotic EC populations are largely absent in these samples, whereas venous, CapV, CapA, and arterial ECs are clearly segregated (*Figure 2A and B* and *Figure 2—figure supplement 3A, B*). Interestingly, we found one additional minor but distinctive subpopulation of venous ECs (*Figure 2A and B*; boxed area), which we subsequently termed REV. These ECs are characterized by the expression of *Icam1* and *Vcam1*, encoding intercellular adhesion molecules (ICAMs), and the endothelial activation or dysfunction markers *Vwf* and *Irf1* (*Figure 2C and D*). It has been reported that endothelial expression of ICAM-1 is essential for transcellular diapedesis (*Abadier et al., 2015*) but is suppressed by sonic hedgehog signalling in the CNS during homeostasis, thereby limiting infiltration of circulating inflammatory leukocytes through the BBB (*Alvarez et al., 2011*). However, the small subpopulation of venous ECs we found from the single-cell transcriptome data shows consistent expression of *Icam1* even in immunologically naïve conditions both in adult and aged mice (*Figure 2D and E*). To further characterize the *Icam1*-expressing REVs, we performed gene ontology (GO) term enrichment analysis and revealed that 663 genes (Padj <0.05) predominantly expressed in this cell population were significantly enriched in biological processes involving inflammatory/immune responses, cell adhesion, and leukocyte migration and adhesion (*Figure 2F*). An alternative, nonlinear dimensionality reduction method called Markov affinity-based graph imputation of cells (MAGIC) (*van Dijk et al., 2018*) also shows a distinct subpopulation of venous ECs with predominant expression of Icam1 and Irf1 (*Figure 2G* and *Figure 2—figure supplement 3C*), excluding the possibility that the heterogeneity of cerebral venous ECs is a computational artifact.

Using immunostaining on 100-μm thick brain vibratome sections from transgenic reporter mice expressing nuclear green fluorescent protein (GFP) and membrane Tomato specifically in ECs (*Cdh5-H2BGFP/tdTomato*) (*Jeong et al., 2017*), we confirmed that ICAM-1[+] REVs are present throughout the cortex of immunologically unchallenged juvenile (*Figure 2—figure supplement 3D*) and adult mouse brain (*Figure 2H*). ICAM-1[+] vessels in the brain parenchyma are predominantly venules with a diameter ranging from 8 to 50 μm (*Figure 2—figure supplement 3E*) and low-to-negative coverage by alpha-smooth muscle actin[+] (αSMA[+]) mural cells (*Figure 2I*), suggesting that they represent a

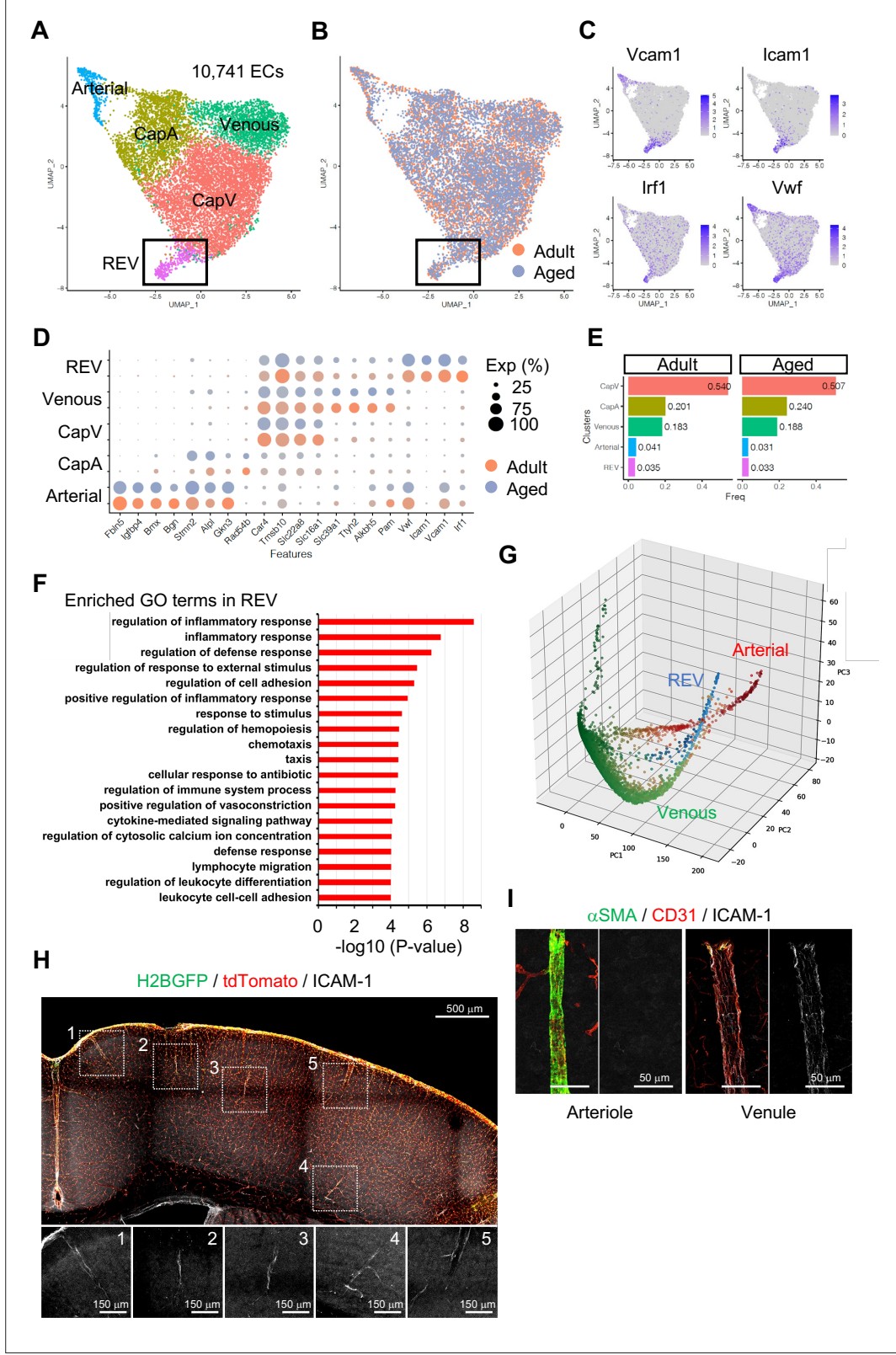

**Figure 2.** Endothelial cell (EC) subclustering and identification of intercellular adhesion molecule-1 (ICAM-1⁺) endothelial population. (**A–B**) Uniform manifold approximation and projection (UMAP) plots of 10,741 adult and aged mouse ECs. Colors represent cell subclusters (**A**) or age groups (**B**), respectively. Box indicates ICAM-1⁺ reactive endothelial venule (REV) ECs. (**C**) UMAP plots depicting the expression of ICAM-1⁺ EC-enriched genes

*Figure 2 continued on next page*

*Figure 2 continued*

*Vcam1*, *Icam1*, *Irf1*, and *Vwf*. Color represents scaled expression level. (**D**) Dot plot showing the expression of top subcluster-specific genes, with the dot size representing the percentage of cells expressing the gene and colors representing the average expression of the gene within a cluster. (**E**) Bar plot showing frequency of subclusters for adult and aged ECs. (**F**) Top gene ontology GO0 biological process terms enriched in REV-specific genes. (**G**) Three-dimensional Principal component analysis (PCA) plots generated by Markov affinity-based graph imputation of cells. Cells are colored representing the expression of selected subtype marker genes (green: *Alkbh5* and *Tmsb10* for venous ECs; red: *Alpl* and *Fbln5* for arterial ECs; blue: *Icam1* and *Vcam1* for REVs). (**H**) Representative immunofluorescence image for ICAM-1 in adult *Cdh5-H2BGFP/tdTomato* murine brain cortex. Scale bar, 500 µm. Panels at the bottom show isolated ICAM-1 signal for each area marked in the overview image, representing different cortical areas. Scale bars, 150 µm. (**I**) Immunostaining showing alpha-smooth muscle actin (αSMA), CD31, and ICAM-1 expression. Panels on the right show ICAM-1 signal for arteriole and venule, respectively. Scale bars, 50 µm.

The online version of this article includes the following source data and figure supplement(s) for figure 2:

**Source data 1.** Source data for *Figure 2E*.

**Figure supplement 1.** Subclustering analysis of endothelial cells (ECs) during postnatal development.

**Figure supplement 2.** Differential gene expression analysis of adult and aged endothelial cells (ECs).

**Figure supplement 3.** Subclustering analysis of endothelial cells (ECs) during homeostasis.

**Figure supplement 3—source data 1.** Source data for panel E.

---

subpopulation of smaller venules, potentially postcapillary venules where leukocyte extravasation into the brain predominantly occurs. While it was not possible to detect VCAM-1 staining in naïve brains, the cellular frequency of *Icam1* and *Vcam1* double-positive ECs ranges from 1 to 3% of total ECs in all stages analyzed (*Figure 2—figure supplement 3F*). Postcapillary venules are the main site of leukocyte extravasation, in most tissues and in the brain, are associated with a perivascular space, defined by the inner endothelial and outer parenchyma or astroglial basement membranes, where activated leukocytes accumulate before entry into the brain parenchyma (*Sixt et al., 2001*; *Zhang et al., 2020*). To test whether the ICAM-1⁺ subpopulation is related to leukocyte trafficking upon an acute inflammation, we intraperitoneally injected adult mice with 10 mg/kg of lipopolysaccharides (LPS), perfused the vasculature with ice-cold PBS after 2 hr to remove nonadherent cells from the vessel lumen, and performed immunostaining. Interestingly, this revealed the accumulation of CD3⁺ lymphocytes at ICAM-1⁺ venules prior to significant upregulation of ICAM-1 in other endothelium (*Figure 2—figure supplement 3G*), suggesting that the ICAM-1⁺ vessel subset might serve as the first entry site of the activated leukocytes to the brain parenchyma in settings of neuroinflammation. Based on these findings, we refer to this vessel compartment as 'reactive endothelial venules' (REVs).

## PVMs reside in close proximity of ICAM-1 expressing REV ECs

Brain parenchyma-resident myeloid cell types, namely microglia and PVMs, share expression of myeloid surface markers including CD68, *Fcgr3* (CD16), Cx3cr1, and Aif1 (*Figure 3—figure supplement 1A*) and are GFP⁺ in *Cx3cr1* reporter mice (*Jung et al., 2000*; *Figure 3—figure supplement 1B*). At the same time, single-cell transcriptome analysis shows molecular differences between the two cell populations (*Figure 3A*), which also correlate with distinct morphologies and localization (*Figure 3—figure supplement 1B, C*). As the heterogeneity of microglia during development, ageing, and brain pathogenesis including Alzheimer's disease, toxic demyelination, and neurodegeneration has been extensively analyzed at the single cell level (*Hammond et al., 2019*; *Keren-Shaul et al., 2017*; *Masuda et al., 2019*), we focused on the characterization of PVMs in this study. CD206 is predominantly expressed by PVMs (*Figure 3—figure supplement 1D*), and CD206⁺ PVMs are localized between the subendothelial basement membrane and GFAP⁺ astrocyte processes (*Figure 3B–D*). PVMs are clearly distinct from EC-associated pericytes (*Figure 3C*, asterisks) and *Pdgfra*-expressing adventitial fibroblast-like cells (*Figure 3D*). Consistent with previous reports showing that PVMs are associated with both arteries and veins in the CNS (*Faraco et al., 2017*; *Faraco et al., 2016*), we found PVMs in the proximity of ICAM-1⁺ vessels in cortex, thalamus, hippocampus, midbrain, and floor plate (*Figure 3E and F*).

Subclustering analysis indicated at least two different PVM populations, namely Lyve1⁺ and MHC class II⁺ (MHCII⁺) PVMs (*Figure 4A–C*), and differential gene expression analysis revealed unique

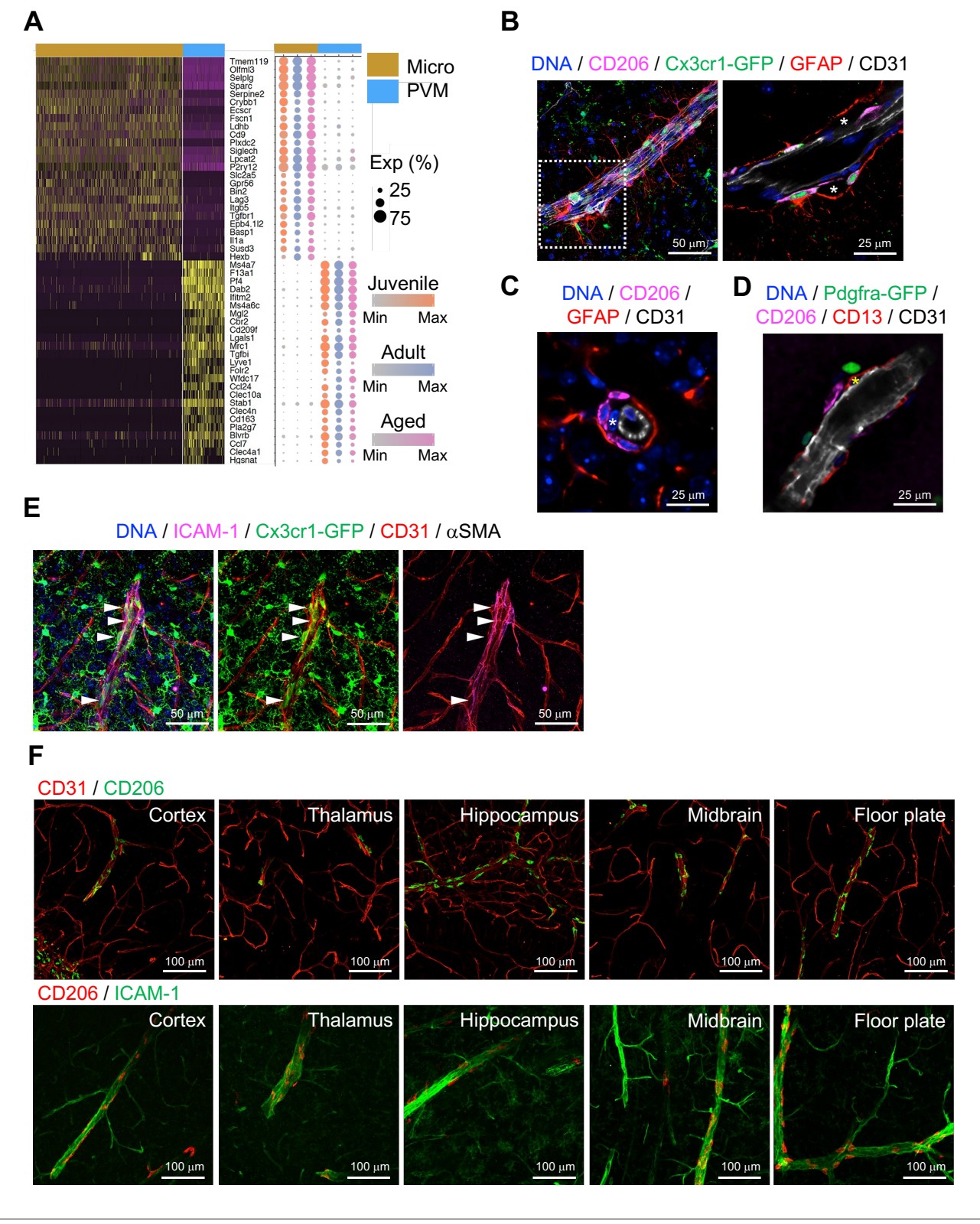

**Figure 3.** Localization of perivascular macrophages (PVMs). (**A**) Heatmap and expression dot plot for top differentially expressed genes between microglia and PVMs. Dot size represents the percentage of cells expressing each gene, and color keys indicate scaled expression in each age group. (**B**) Immunostaining showing CD206⁺ *Cx3cr1-GFP*⁺ PVMs in perivascular space (asterisk) between ECs (CD31) and astrocyte limitans (GFAP). Scale bars, 50 μm (left), 25 μm (right). (**C–D**) Spatial arrangement of PVMs (CD206), astrocyte limitans (GFAP), pericytes (CD13, asterisks), perivascular fibroblasts

*Figure 3 continued on next page*

*Figure 3 continued*

(*Pdfgra- GFP⁺*), and endothelial cells (ECs; CD31). Scale bars, 25 µm. (**E**) Confocal image showing *Cx3cr1-GFP* ⁺ microglia and vessel-associated PVMs (arrowheads) next to ICAM1⁺αSMA⁻ CD31⁺ REV ECs. Scale bar, 50 µm. (**F**) CD206⁺ PVMs are associated with ICAM-1⁺ REVs in the indicated brain regions. Scale bars, 100 µm.

The online version of this article includes the following figure supplement(s) for figure 3:

**Figure supplement 1.** Distinct populations of microglia and perivascular macrophages (PVMs).

transcriptomic landscapes of the two subtypes (*Figure 4D*, *Figure 4—figure supplement 1A, B*). Immunostaining of Lyve1 and MHCII also shows the two subtypes in association with ICAM-1⁺ vessels (*Figure 4E*). Interestingly, GSEA for KEGG signalling pathways indicates that Lyve1⁺ PVMs show enriched expression of genes associated with lysosome activity and endocytosis as well as Wnt signalling, whereas MHCII⁺ PVMs show enrichment of terms such as antigen processing and presentation, cell adhesion molecules, and Toll-like receptor signalling pathway (*Figure 4F*, *Figure 4—figure supplement 1C*). Macrophages can be broadly divided into classical (M1) and alternative (M2) subtypes (*Mantovani et al., 2005*; *Mantovani et al., 2004*). M1-polarized macrophages are strongly positive for MHC class II and present antigen to T lymphocytes that neutralize cells with viral or bacterial infections. Macrophages with M2 polarization display homeostatic or anti-inflammatory activity and have a higher endocytic ability compared to M1 macrophages (*Tarique et al., 2015*). Interestingly, Lyve1⁺ PVMs express anti-inflammatory or immune-suppressive (M2 macrophage-like) polarization genes, such as *Ccr1*, *Cd163*, *Cd209a/f*, *Cd302*, *Igf1*, *Il21r*, *Mrc1*, *Stab1*, *Tgfb1*, and *Tslp*, while MHCII⁺ PVMs show increased levels of proinflammatory (M1-like) genes, *Cxcl9*, *Cxcl10*, *Cxcl13*, *Cxcl16*, *Irf5*, *Il1a/b*, *Cxcr4*, *Il2ra*, and *Tlr2* (*Figure 4G*). Both PVM subtypes are found at all stages investigated, but the frequency of MHCII⁺ PVMs increases with ageing, which correlates with a decrease in the frequency of Lyve1⁺ PVMs (*Figure 4H*). Not only the cell population but also the expression levels of MHCII genes are upregulated, whereas *Lyve1* or *Ccl24* expression is downregulated in aged PVMs (*Figure 4I*). These results indicate that specific PVM subtypes, characterized by distinct immunological signatures, are associated with ICAM-1⁺ REVs.

## ICAM-1⁺ ECs are the most reactive EC population in EAE

To extend our analysis of brain ECs and PVMs to neuroinflammatory disease, we analyzed the transcriptome of non-neuronal cell types in the brain cortex during EAE at single-cell resolution. In contrast to age-matched control mice, brain-infiltrating inflammatory cells, activated macrophages, neutrophils, and lymphocytes were significantly enriched in EAE (*Figure 5—figure supplement 1A, B*). EAE mice show onset of disease symptoms between 9 and 12 days and peak disease severity between 14 and 18 days after immunization with significant reduction of myelin basic protein (MBP) levels in the brain cortex (*Figure 5—figure supplement 1C*). The comparative analysis of ECs from healthy adults vs. peak EAE indicates the emergence of inflammatory ECs upon neuroinflammation (*Figure 5A*, *Figure 5—figure supplement 2A*). These ECs share the molecular features of CapV (*Figure 5—figure supplement 2B*) but show higher level of genes related to immune responses such as *Ctla2*, *Lcn2*, and *Mt2* (*Figure 5—figure supplement 2C*). REV-specific marker genes, *Icam1* and *Lrg1* are also upregulated in inflammatory ECs (*Figure 5B*). Consistent with published reports (*Dopp et al., 1994*), both the area and level of endothelial ICAM-1 expression are increased in EAE (*Figure 5C and D*). Interestingly, ICAM-1 expression in the brain parenchyma persists in REVs and, in peak EAE, expands into adjacent vessels (*Figure 5C*, *Figure 5—figure supplement 3A*).

It has been reported that high levels of ICAM-1 expression in ECs promote transcellular diapedesis of encephalitogenic T cells and, in absence of endothelial ICAM-1 and ICAM-2 in mutant mice, EAE symptoms were ameliorated (*Abadier et al., 2015*). Accordingly, upon EAE, leukocytes accumulate at ICAM-1⁺ REVs and enter the adjacent brain parenchyma (*Figure 5E*, *Figure 5—figure supplement 3B, C*).

EAE alters gene expression in REVs (*Figure 5—figure supplement 4A, B*), including significant upregulation of regulators of blood vessel development, such as *Pdgfb* and *Ctgf*, *Cd74* antigen (invariant polypeptide of the major histocompatibility complex, class II antigen-associated), the hemostasis regulator *Vwf*, an interferon gamma-stimulated gene mediating host immune responses (*Ch25h*), a gene involved in the cellular stress response (*Maff*), and numerous other genes related to antigen processing and presentation as well as cellular responses to hydrogen peroxide or radiation

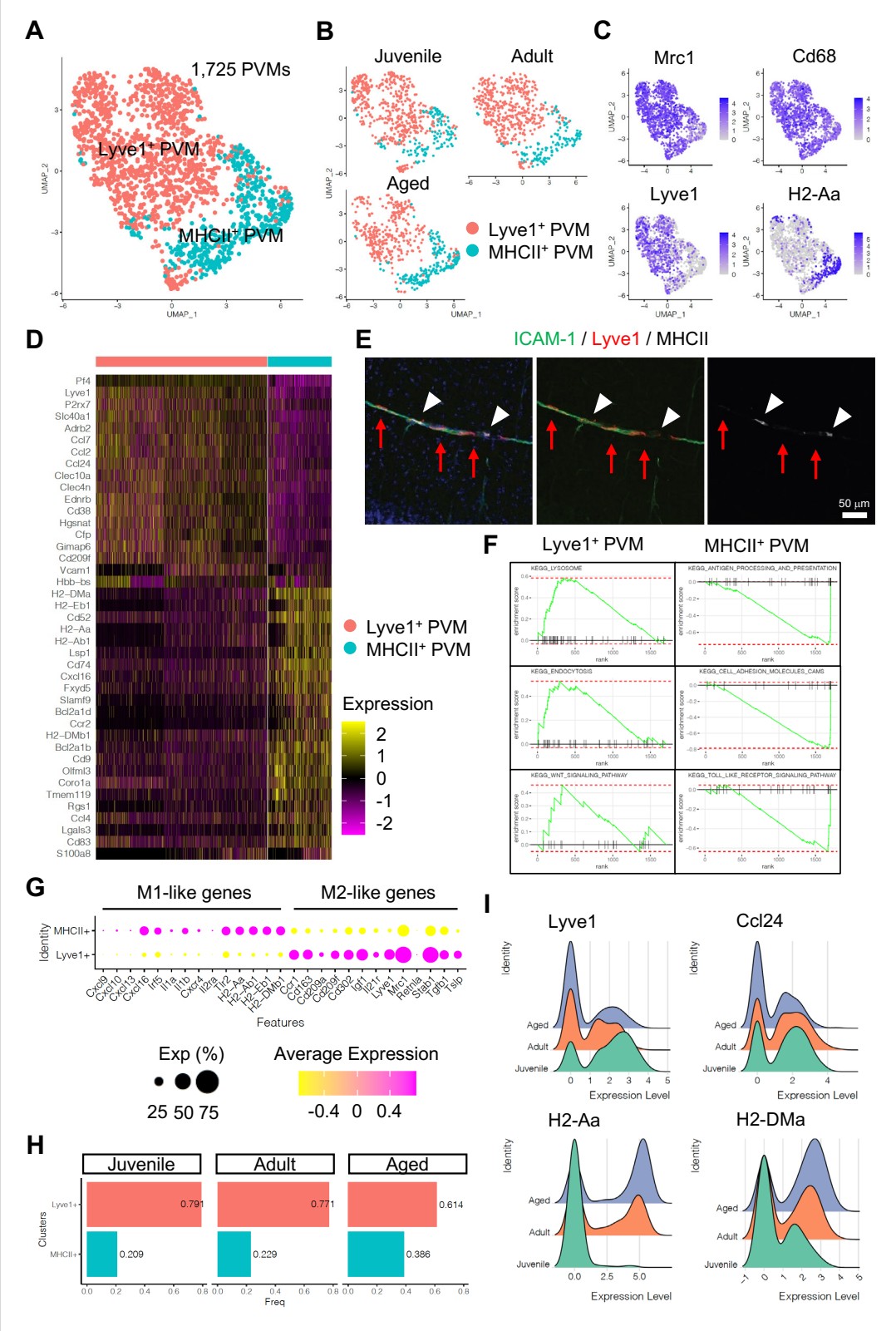

**Figure 4.** Perivascular macrophage (PVM) heterogeneity. (**A**) Uniform manifold approximation and projection (UMAP) plot of 1725 PVMs from juvenile, adult, and aged mice distributed into Lyve1+ and MHCII+ subclusters. (**B**) Split UMAP plots of PVMs from juvenile, adult, and aged samples. (**C**) UMAP plots depicting the expression of *Mrc1*, *Cd68*, *Lyve1*, and *H2-Aa*. Color represents the scaled expression level. (**D**) Heatmap of top marker genes for Lyve1+ and MHCII+ PVMs. (**E**) Immunostaining showing Lyve1+ (red arrows) or MHCII+ PVMs (arrow heads) associated to ICAM-1+ REVs. Scale bar, 50 μm.

*Figure 4 continued on next page*

*Figure 4 continued*

(**F**) Representative gene set enrichment analysis plots for overrepresented KEGG pathways in Lyve1⁺ and MHCII⁺ PVMs. (**G**) Dot plot of genes related to M1- or M2-like phenotypes in PVM subtypes. Dot size represents percentage of cells expressing the gene, and colors represent the average expression of each gene. (**H**) Bar plot showing the frequency of PVM subtypes in different age groups. (**I**) Ridge plots of *Lyve1*, *Ccl24*, and MHCII genes in PVMs at different ages.

The online version of this article includes the following source data and figure supplement(s) for figure 4:

**Source data 1.** Source data for *Figure 4H*.

**Figure supplement 1.** Differentially expressed genes in perivascular macrophage (PVM) subtypes.

(*Figure 5—figure supplement 4C, D*). Collectively, our results indicate that ICAM-1⁺ postcapillary venules or REVs are molecularly specialized and function as a gateway for the entry of activated leukocytes through the BBB.

## PVMs in EAE and cell-to-cell interactome analysis of BBB

The analysis of PVMs in adult and EAE mice indicates that these cells rarely proliferate (*Figure 6—figure supplement 1A*), and their morphology and localization around ICAM-1⁺ REVs persist in EAE (*Figure 6A and B* and *Figure 6—figure supplement 1B*). Nevertheless, EAE induces profound changes in PVM gene expression. MHCII⁺ PVM-enriched genes, such as *Cxcl16*, *Ccl8*, *Cd52*, and MHCII genes, are significantly upregulated in most of the PVMs, whereas Lyve1⁺ PVM-enriched genes, namely *Lyve1*, *Igf1*, *Cd163*, *Ccl24*, *Folr1*, and *Cd209f*, are not changed or decreased (*Figure 6D and E*). This supports the hypothesis that polarization of macrophages toward inflammatory and non-inflammatory phenotypes is not fixed, and that these cells possess plasticity, integrating diverse inflammatory signals with physiological and pathological functions (*Jordão et al., 2019*; *Murray, 2017*). In addition, differential gene expression analysis comparing PVMs and brain parenchyma-infiltrated activated macrophages (Mac) further illustrates the molecular differences between PVMs and circulating peripheral monocytes/macrophages (*Figure 6F and G*), which may prove relevant for potential therapeutic approaches involving the targeting and modulation of different types of macrophages.

We next used CellPhoneDB (*Vento-Tormo et al., 2018*) to identify potential receptor-ligand complexes mediating cell-to-cell communication between different cell populations at the BBB. First, we analyzed the communication between REVs and adjacent CapV and venous ECs. REVs show expression of various autologous signalling molecules regulating fundamental aspects of EC behavior, including *Wnt5a*, *Tnfsf12*, *Tgfb1*, *Jag1*, collagens, and immune-regulatory genes with their corresponding receptors (*Figure 7A*). CapV and venous ECs express higher level of *Esam*, which encodes EC-selective adhesion molecule, an immunoglobulin-like transmembrane protein associated with endothelial tight junctions. CapV and venous ECs also express the chemokine, and *Cxcl12* and REVs express the corresponding receptor *Ackr3* encoding CXCR-7, which might enable communication between EC subsets. Potential molecular interactions that upregulated in REVs by EAE include members of the tumor necrosis factor receptor superfamily, TGFβ receptors, integrins, the HLA class II histocompatibility antigen gamma chain CD74, the receptor tyrosine kinase TEK, and the PLXNB2 receptor for semaphorin ligands (*Figure 7A and B*). These results also suggest that REVs are a major signal distributor among the EC populations in CNS, which is further enhanced by EAE (*Figure 7C*). We also identify putative signalling interactions between REVs and other cell types in the brain vasculature during homeostasis and neuroinflammation (*Figure 7—figure supplement 1*). Notably, REVs express growth factors including PDGF-B, CTGF, and TGFβ1, which regulate cellular behavior, tissue remodelling, and angiogenesis. Signalling to other cell types is further enhanced by EAE, which involves the activation of various signals in astroependymal cells. By contrast, expression of PDGF-A, IGF2, and PGF in mural cells and of IGF1, TGFB1, and IGFBP4 in PVMs, for example, was not significantly changed upon EAE (*Figure 7D*). Our results collectively suggest that postcapillary venules form a specialized vessel compartment in the CNS. We propose that REVs might play roles in the regulation of immune surveillance in the CNS not only under homeostatic conditions but also in pathogenic neuroinflammation.

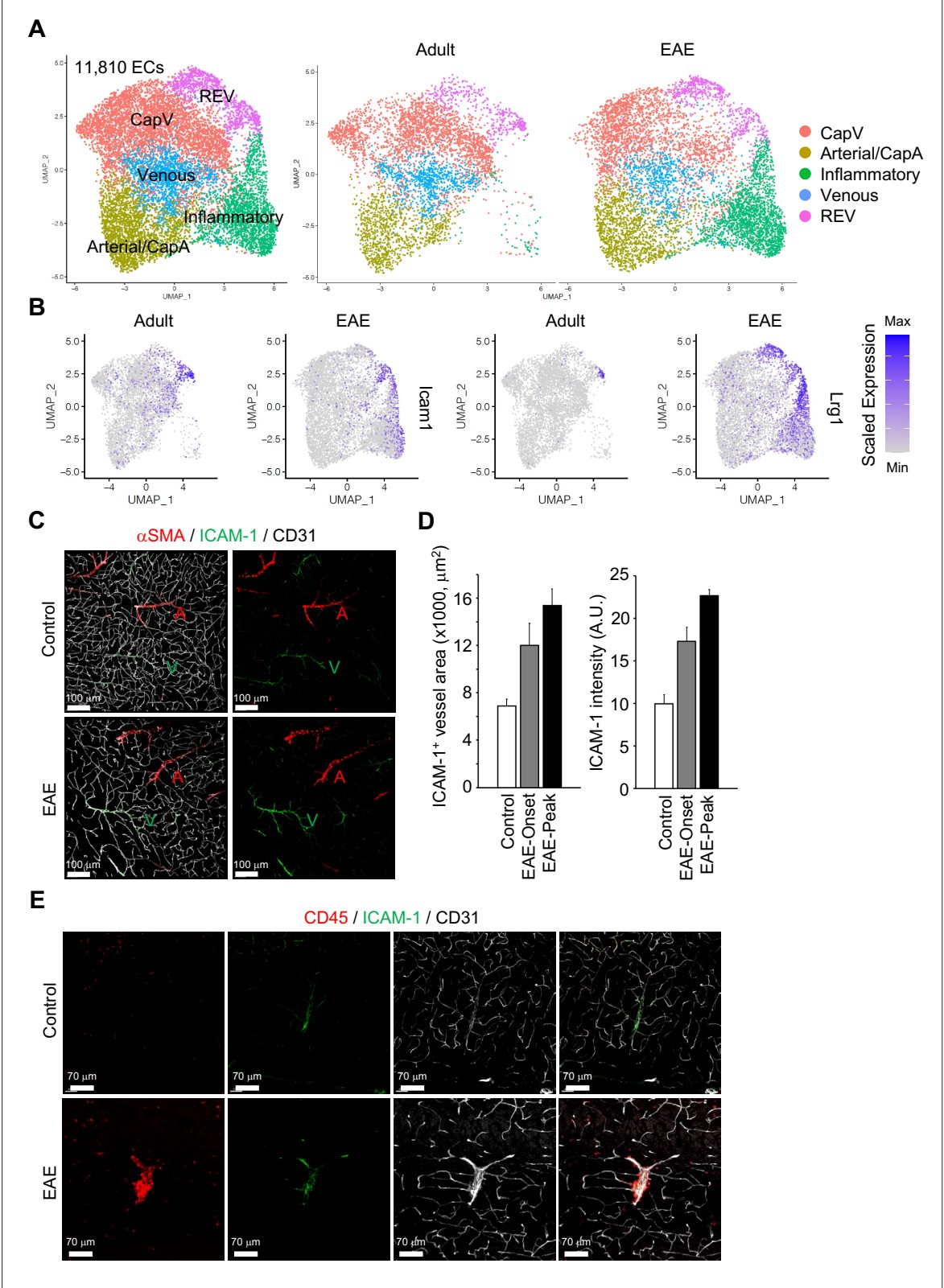

**Figure 5.** Gene expression changes in endothelial cells (ECs) by experimental autoimmune encephalomyelitis (EAE). (**A**) Uniform manifold approximation and projection (UMAP) plots of 11,810 ECs from adult and EAE mice. Colors represent each EC subcluster (left) or condition (right). (**B**) UMAP plots depict *Icam1* and *Lrg1* expression in ECs from adult and EAE mice. Color represents the scaled expression level. (**C**) Immunostaining showingαSMA, CD31, and ICAM-1 expression in control and EAE mice. Arterial (**A**) and venous vessels (**V**) are indicated. Scale bars, 100 μm. (**D**)

*Figure 5 continued on next page*

*Figure 5 continued*

Quantification of ICAM-1$^+$ vessel area and immunohistochemistry signal intensity in the brain cortex of control and EAE mice at disease onset (EAE onset) and peak (EAE peak). Error bars represent mean ± s.e.m. from three animals. (**E**) Immunofluorescence images showing CD45$^+$ leukocytes near ICAM-1$^+$ REVs in the cerebral cortex of control and EAE mouse. CD31 indicates all ECs. Scale bars, 70 μm.

The online version of this article includes the following source data and figure supplement(s) for figure 5:

**Source data 1.** Source data for *Figure 5D*.

**Figure supplement 1.** Single-cell RNA-seq analysis of experimental autoimmune encephalomyelitis (EAE) mice.

**Figure supplement 2.** Subclustering of endothelial cells (ECs) from adult and experimental autoimmune encephalomyelitis (EAE) mice.

**Figure supplement 2—source data 1.** Source data for panel A.

**Figure supplement 3.** Intercellular adhesion molecule-1 (ICAM-1) expression in brain cortical parenchyma upon experimental autoimmune encephalomyelitis (EAE).

**Figure supplement 4.** Differentially expressed genes between adult and experimental autoimmune encephalomyelitis (EAE) endothelial cells (ECs).

## Discussion

Here, we provide a comprehensive single-cell transcriptomic atlas of non-neural components of the murine brain vasculature during growth, adulthood, ageing, and in EAE. Using this data, we establish the existence of a specialized vessel subtype, which is characterized by a distinct endothelial gene expression profile and presence of ICAMs. In response to pro-inflammatory (LPS) stimulation or in the neuroinflammatory condition of EAE, REVs, but not the far more abundant ICAM-1$^-$ vessels, are associated with leukocytes, suggesting that the ICAM-1$^+$ subset of postcapillary venules serves as a first site of immune cell accumulation prior to entry into the brain parenchyma. Such a role of REVs as an interface for innate and adaptive immune cells might be also relevant under homeostatic conditions, when the recruitment of these cells is rare and their activity but also their entry into the brain parenchyma is tightly regulated (*Schwartz et al., 2013*). The association of MHCII$^+$ and Lyve1$^+$ PVM populations with REVs might mediate immunomodulatory responses, and the predominant gene expression changes in REVs upon immunological challenge highlight further that these vessels might be critical for the immune privileged status of the CNS with potential implications for ageing and neurodegenerative diseases.

Several previous studies on scRNA-seq analysis of murine or human brain have recovered only a limited number of vascular cells due to the comparably high abundance of neural cells (*Han et al., 2020*; *Zeisel et al., 2015*). In other publications, fluorescence-activated cell sorting (FACS) or microbead-mediated isolation of ECs as well as of both ECs and mural cells has provided insight into the organ-specific specialization, transcriptional regulation, and arterial-venous zonation of brain vessels in the healthy organism (*He et al., 2018*; *Kalucka et al., 2020*; *Sabbagh et al., 2018*; *Vanlandewijck et al., 2018*). While sorting based on certain cell surface markers or fluorescent reporters enables the efficient enrichment of the desired cell populations, it has to be considered that these approaches are biased and will miss cells lacking expression of the relevant markers. This limitation is avoided by our demyelination approach, which does not rely on the expression of specific cell surface proteins or fluorescent reporters. Despite of the differences in the experimental approach, data generated by *Vanlandewijck et al., 2018* confirms the enriched expression of Icam1 and Vwf in a fraction of venous ECs, which is consistent with our own findings. Transcriptomic changes in brain ECs associated with the aging have been also investigated previously, and it has been proposed that soluble brain EC-derived Vcam1, generated by shedding, activates microglia and impairs the function of hippocampal neural precursors (*Chen et al., 2020*; *Yousef et al., 2019*). Our own study confirms that aging increases transcripts associated with inflammation in brain ECs, and, in particular, in REVs. Similarly, EAE results in the upregulation of proinflammatory gene expression in ECs with potential implications for neuroinflammation and disease development.

In the healthy organism, the CNS contains a highly selective barrier system to restrict peripheral immune cell entry into brain parenchyma. During pathophysiological conditions, activated T cells can enter the perivascular space independent of antigen specificity (*Kawakami et al., 2005*) along postcapillary venules (*Raine et al., 1990*). However, only after acquiring the ability to move across the glia limitans by recognition of cognate antigen displayed by perivascular APCs, T cells traverse into the parenchyma (*Lodygin et al., 2013*). Firm attachment of T cells to ECs to resist the vascular shear

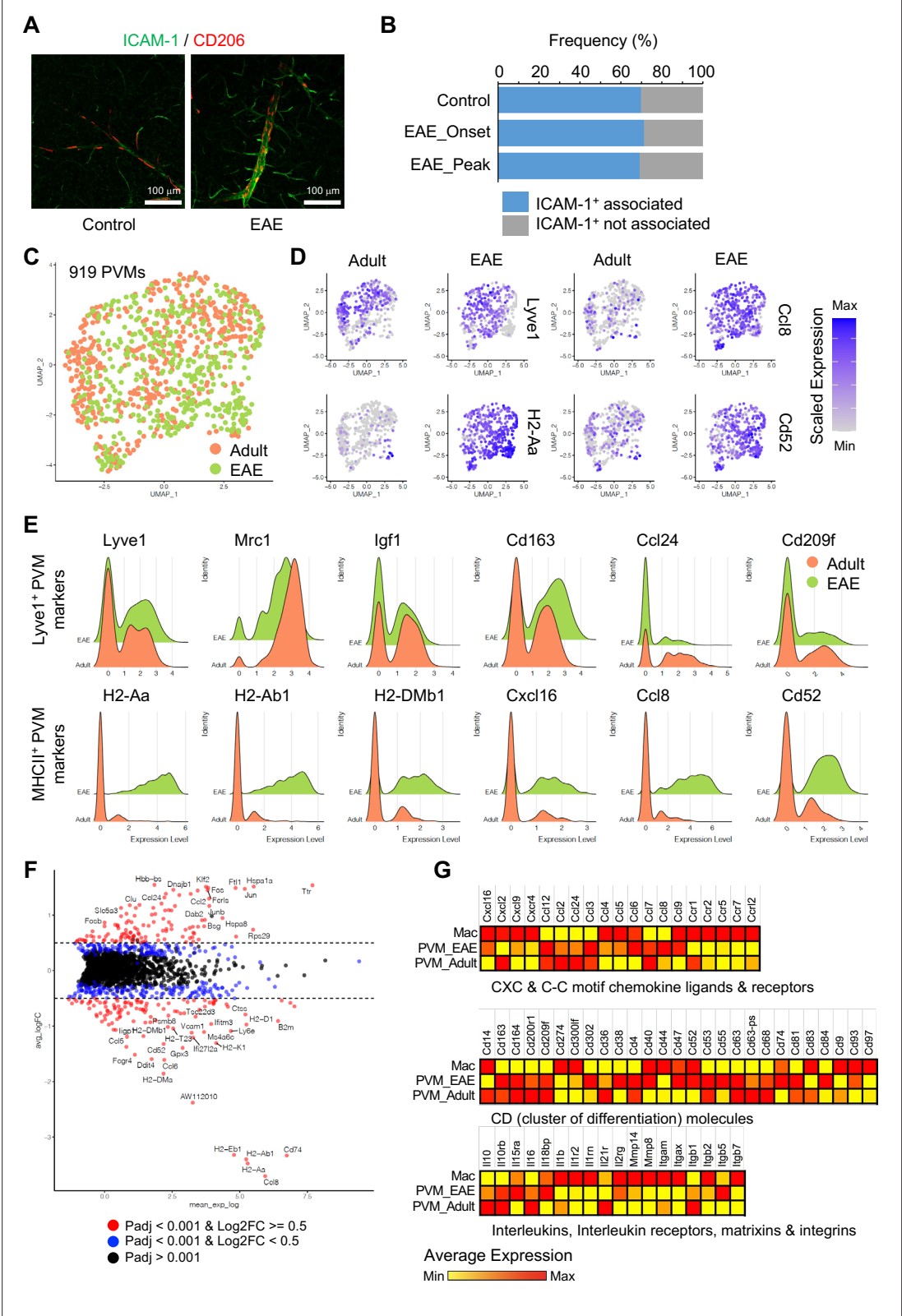

**Figure 6.** Experimental autoimmune encephalomyelitis (EAE)-induced changes in perivascular macrophages (PVMs). (**A**) Representative immunohistochemistry images for CD206+ PVMs and ICAM-1+ vessels in control and EAE brain cortical parenchyma. Scale bars, 100 μm. (**B**) Frequency of CD206+ PVMs associated to ICAM-1+ vessels in brain cortical parenchyma of control and EAE mice at disease onset (EAE onset) and peak (EAE peak). (**C**) Uniform manifold approximation and projection (UMAP) plots of 919 PVMs from adult and EAE mice. (**D**) UMAP plots depicting the expression

*Figure 6 continued on next page*

*Figure 6 continued*

of *Lyve1*, *H2-Aa*, *Ccl8,* and *Cd52* in PVMs of adult control and EAE mice. Color indicates the scaled expression. (**E**) Ridge plots of Lyve1⁺ and MHCII⁺ PVM marker genes in adult and EAE PVMs. (**F**) MA plot of differentially expressed genes between PVMs from healthy adult and EAE brain. Blue dots, p-adjusted value <0.001; red dots, p-adjusted value <0.001, and Log2 fold change >0.5. (**G**) Heatmap of selected differentially expressed genes (Padj <0.001) across adult and EAE PVMs and activated macrophages (Mac). Color indicates the scaled average expression.

The online version of this article includes the following source data and figure supplement(s) for figure 6:

**Source data 1.** Source data for *Figure 6B*.

**Figure supplement 1.** Cell cycle phases and localization of perivascular macrophages (PVMs) in experimental autoimmune encephalomyelitis (EAE).

stress and prolonged crawling before diapedesis is also critical for immune cell extravasation (*Steiner et al., 2010*). Our findings suggest that functionally specialized REV ECs enable the sustained immune surveillance process alongside the postcapillary venules, serving as the cellular gateway of BBB. In this sense, REVs share some functional features with high endothelial venules in secondary lymphoid organs that support circulating lymphocyte extravasation and physical contact with APCs (*Girard and Springer, 1995*). Because of the restricted afferent and efferent communication with lymphatic tissue in CNS parenchyma, these specialized postcapillary ECs are probably relevant for fine-tuned CNS immune reactions.

Our work also shows transcriptionally distinguishable subtypes of brain resident PVMs, physically associated with REVs in postcapillary venules, which might serve as APCs for CNS immune surveillance and thereby as gate-keepers at the BBB. It has been reported that PVMs are derived from early yolk-sac-derived erythromyeloid progenitors, similar to microglia and have minimal turnover during homeostasis (*Goldmann et al., 2016*). The transcriptional continuum between the PVM subtypes and the coherent activation upon EAE reflect the ability of PVMs to swiftly adapt to changing immunological and environmental cues. Nevertheless, PVMs retain distinct molecular characteristics relative to infiltrating peripheral macrophages even though all these cells are exposed to the same perivascular microenvironment. More detailed understanding of the mechanisms that determine the behavior of PVMs and peripheral macrophages upon neuroinflammation will be important, especially for attempts to target specific cell populations and modulate local immune responses. Delivery of drugs or therapeutic cells (e.g. chimeric antigen receptor T cells) across the BBB is a limiting factor in the future development of new therapeutics for the brain diseases, such as Alzheimer's disease and various brain tumours (*Pardridge, 2019*).

The sum of our work reveals critical aspects of vascular heterogeneity in the CNS and thereby provides a valuable resource for cell-cell interactions and the molecular modulation of immune cell trafficking into the CNS, with relevance for brain function in health and disease.

## Methods

### Mice

C57BL/6 female mice were used unless stated otherwise. *Cdh5- H2BGFP/tdTomato* (*Jeong et al., 2017*), *Cx3cr1-GFP* (*Jung et al., 2000*), and *Pdgfra-GFP* transgenic mice were used to specifically label EC, Micro + PVM, and Fibro, respectively. All animal experiments were performed in compliance with the relevant laws and institutional guidelines, approved by local animal ethic committees, and conducted with permissions (84–02.04.2016 .A525, 81–02.04.2020 .A471 and 84–02.04.2017 . A322) granted by the Landesamt für Natur, Umwelt und Verbraucherschutz (LANUV) of North Rhine-Westphalia, Germany.

### EAE and LPS treatment

EAE was induced as previously described (*Gerwien et al., 2016*). In brief, 150 µl of 0.75 mg/ml myelin oligodendrocyte glycoprotein 33–55 peptides (MOG35–55) mixed with Complete Freund's Adjuvant (FS8810; Sigma-Aldrich) was injected subcutaneously into the tail base of 7–11-week-old female C57Bl/6 mice. On the day of immunization and on day 2, 100 µl of 2 µg/ml pertussis toxin (P7208; Sigma-Aldrich) was injected intravenously. Mice were monitored daily for development of disease symptoms. EAE was graded on a 0–5 scale as follows: score 0, no clinical symptoms; score 1, flaccid tail; score 2, hindlimb weakness or partial paralysis; score 3, severe hindlimb weakness or paralysis;

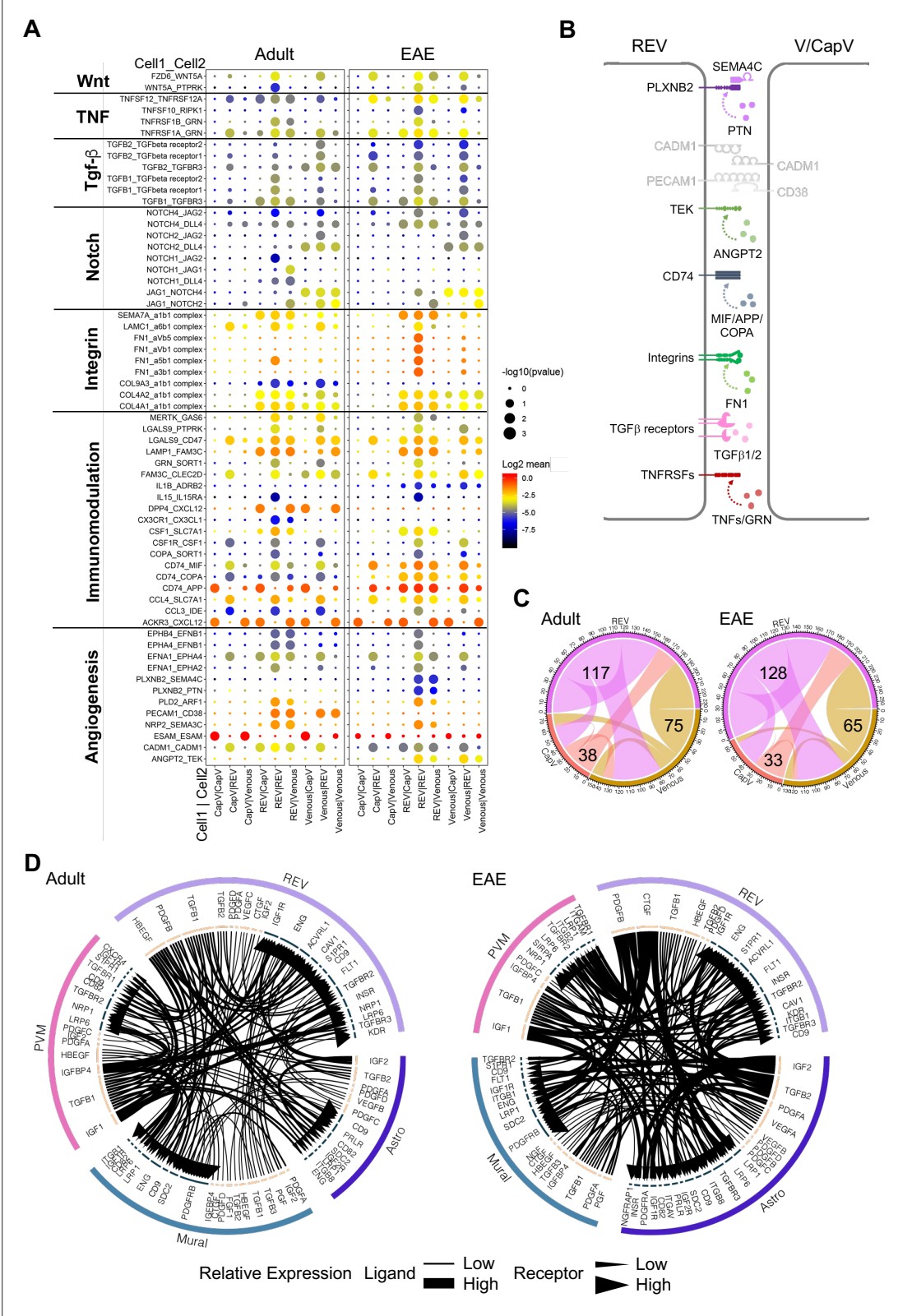

**Figure 7.** Interactions between different endothelial cell (EC) subpopulations and blood-brain barrier (BBB) cell types. (**A**) Overview of potential ligand-receptor interactions for CapV, reactive endothelial venule (REV), and venous EC populations in adult and experimental autoimmune encephalomyelitis (EAE) based on gene expression. Circle size indicates p-values. Color key indicates the means of the average expression levels of interacting molecules. (**B**) Diagram of the main ligand-receptor interactions regulated by EAE. (**C**) Diagram of the numbers of ligand-receptor interactions on CapV, REV,

*Figure 7 continued on next page*

*Figure 7 continued*

and venous EC populations in healthy adult and EAE brain. (**D**) Circos plots for ligand-receptor interactions between REV ECs, mural cells (Mural), perivascular macrophages (PVMs), and astroependymal cells (Astro) in adult and EAE. Each plot shows top 100 highly expressed interactions. The lines and arrow heads are scaled to indicate the relative expression level of the ligand and receptor, respectively.

The online version of this article includes the following figure supplement(s) for figure 7:

**Figure supplement 1.** Interactome of blood-brain barrier (BBB) cell types in postcapillary venules.

**Figure supplement 2.** Schematic summary of experimental findings.

score 4, forelimb paralysis; and score 5, paralysis in all limbs or death. Mice typically reached a peak of disease symptoms between days 14 and 18 after immunization, before disease resolution. The disease typically correlates with weight loss; killing of mice was required when weight loss exceeded 20–30% of the initial body weight.

For acute LPS treatment, mice were intraperitoneally injected with a single dose of 10 mg/kg LPS (L2630-25MG; Sigma-Aldrich). Control animals received the same volume of PBS (data not shown). Mice were sacrificed at 2 hr after the injection.

## Single-cell RNA-seq library preparation and sequencing

Mice were perfused transcardially with ice-cold PBS under anesthesia. Brains were isolated and dabbed with filter paper for the removal of meninges. After the removal of olfactory bulbs and cerebellum, brain cortex tissues were dissected and digested using the enzyme-cocktail solution of 30 U/ml papain (LK003153; Worthington) and 0.125 mg/ml Liberase TM (5401119001; Sigma-Aldrich) in Dulbecco's Modified Eagle Medium (DMEM, Sigma-Aldrich) for 60 min at 37°C. Homogenized tissues were combined with 1.7-fold volume of 22% BSA in PBS and then centrifuged at 1000× g for 10 min for myelin removal. After washing with ice-cold DMEM, dead cells and debris were removed using ClioCell nanoparticles (Amsbio) according to the manufacturer's instructions. Single cells were counted using Luna-II automated cell counter (Logos Biosystems) and captured using the 10X Chromium system (10X Genomics). We pooled tissues from three (for aged and EAE samples) or six (for juvenile and adult samples) mice to reduce the effect of sample-dependent individual variations. Libraries were prepared according to the manufacturer's instructions using Chromium Single Cell 3' Library & Gel Bead Kit v2 (10X Genomics) and sequenced on the Illumina NextSeq 500 using High Output Kit v2.5 (150 cycles, Illumina) for 26 bp +98 bp paired-end reads with 8 bp single-index aiming raw sequencing depth of >20,000 reads per cell for each sample.

## Single-cell RNA-seq data analysis

Sequencing data were processed with UMI tools (*Smith et al., 2017*) (version 1.0.0), aligned to the mouse reference genome (mm10) with STAR (*Dobin et al., 2013*) (version 2.7.1 a), and quantified with Subread featureCounts (*Liao et al., 2014*) (version 1.6.4). Data normalization, detailed analysis, and visualization were performed using Seurat package (*Butler et al., 2018*) (version 3.1.5). For initial quality control of the extracted gene-cell matrices, we filtered cells with parameters nFeature_RNA >500 & nFeature_RNA <6000 for number of genes per cell and percent.mito <25 for percentage of mitochondrial genes and genes with parameter min.cell=3. Filtered matrices were normalized by LogNormalize method with scale factor = 10,000. Variable genes were found by FindVariableFeatures function with parameters of selection.method = 'vst', nfeatures = 2000, trimmed for the genes related to cell cycle (GO:0007049) and then used for principal component analysis. FindIntegrationAnchors and IntegrateData with default options were used for the data integration. Statistically significant principal components were determined by JackStraw method, and the first 12 principal components were used for UMAP nonlinear dimensional reduction. Unsupervised hierarchical clustering analysis was performed using FindClusters function in Seurat package. We tested different resolutions between 0.1 and 0.9 and selected the final resolution using clustree R package to decide the most stable as well as the most relevant for our previous knowledges. Cellular identity of each cluster was determined by finding cluster-specific marker genes using FindAllMarkers function with minimum fraction of cells expressing the gene over 25% (min.pct=0.25), comparing those markers to known cell type-specific genes from previous studies.

Data were further trimmed for clusters of multiplets, low-quality cells (mitochondrial gene-enriched), contaminated neurons (Tubb3+), oligodendrocytes (Mbp+), lymphocytes (Igkc+ or Nkg7+), red blood cells (Hba−a1+), and ependymal cells (Tmem212+) and then reanalyzed. For subclustering analysis, we isolated specific cluster(s) using subset function, extracted data matrix from the Seurat objectusing GetAssayData function, and repeated the whole analysis pipeline from data normalization.

Differentially expressed genes were identified using the nonparametric Wilcoxon rank sum test by FindMarkers function of Seurat package. We used default options for the analysis if not specified otherwise. Results were visualized using EnhancedVolcanoR package (version 1.10.0). FeaturePlot, VlnPlot, and DotPlot functions of Seurat package were used for visualization of selected genes. The 'VlnPlot' function of Seurat package was used for violin plots to show the expression level of selected genes with log normalized value by default.

Monocle (*Trapnell et al., 2014*) (version 2.12.0) was used for pseudotime trajectory analysis. We imported Seurat objects to Monocle R package and then performed dimensionality reduction with DDRTree method with parameters max_components = 2 and norm_method='log'. Cell cycle phases were classified by cyclone function of scran (*Lun et al., 2016*) (version 1.14.5). We also used MAGIC (*van Dijk et al., 2018*) (version 1.4.0.9000) for an alternative dimensionality reduction and Cell-PhoneDB (*Vento-Tormo et al., 2018*) for ligand-receptor interactome analysis. An R package iTALK (https://doi.org/10.1101/507871) was used for the ligand-receptor interactome analysis. For the analysis of each single sample, top 50% of genes in their mean expression values were selected and used for ligand-receptor pair identification using FindLR function with datatype = mean count. All the analysis scripts are available from vignettes of original software webpage of Seurat, Monocle, MAGIC, CellPhoneDB, or iTALK. No custom code or mathematical algorithm other than variable assignment was used in this study.

## Flow cytometry

Brain cortex tissues from transcardially perfused P10 *Cdh5- H2BGFP/tdTomato* transgenic mice were dissected and treated by different digestion conditions, namely papain solution, papain + liberase cocktail solution, or papain + liberase cocktail solution followed by myelin removal. EC-specific H2BGFP and tdTomato fluorescences were directly analyzed after single/live cell gating in forward scattered/side scattered plot using FACSVerse flow cytometer (BD). For brain-infiltrated leukocyte analysis, brains from transcardially perfused mice were dissected and crushed with FACS buffer (2% FCS, 2 mM EDTA in DPBS) using 100 um mesh and plunger. After collecting the cell suspension with 10 ml of FACS buffer, centrifugation was performed to make cell pellet. Percoll gradient was performed, and a layer that contains mononuclear cells was collected for further FACS analysis. $10^5$ cells were stained with fluorescence conjugated antibodies, CD45-APC-Cy7 (BD 557659) and CD3e-FITC (Thermo 11–0031) for 30 min at 4°C. The analysis is performed with BD FACSymphony A3 Cell Analyzer. FlowJo software (version 10.5.3, FlowJo, LLC) was used for further analysis.

## Immunohistochemistry

Mice were perfused transcardially with ice-cold PBS and subsequently with 2% paraformaldehyde (PFA) under anesthesia. Whole brain tissues were dissected and further fixed with 4% PFA or 100% methanol at 4°C for overnight. Methanol-fixed samples were rehydrated by serial incubation (15–20 min each, at RT) in increasing concentrations of PBS:methanol solution (25, 50, 75, and 100% PBS). The fixed brains were glued to a mounting block with cyanoacrylate glue (48700; UHU), submerged in ice-cold PBS, and sliced with 100-μm thickness using vibrating blade microtome (VT1200, Leica). Sections were blocked and permeablilized by 1% BSA and 0.5% Triton X-100 in PBS for overnight. Incubation with blocking/permeabilization solution containing primary antibodies at 4°C for overnight was followed by secondary antibody staining using suitable species-specific Alexa Fluor-coupled antibodies (Invitrogen) and flat-mounting in microscope glass slides with Fluoromount-G (0100–01; SouthernBiotech). The following primary antibodies were used for immunostaining: rabbit anti-ICAM-1 (Abcam ab222736, 1:100), rat anti-ICAM-1 (BioLegend 116102, 1:100), mouse anti-αSMA-Cy3 (Sigma C6198, 1:300), mouse anti-αSMA-660 (eBioscience 50-9760-82, 1:300), chicken anti-GFP (2BScientific Ltd. GFP-1010, 1:300), goat anti CD31 (R&D Systems AF3628, 1:200), rabbit anti-CD206 (Abcam ab64693, 1:100), rat anti-CD206 (BioRad MCA2235T, 1:100), rabbit anti-GFAP (DAO Z0334, 1:200), rat anti-CD13 (AbD Serotec MCA2183GA, 1:100), hamster anti-CD3e-FITC (eBiosciences 11–0031, 1:100), rabbit anti

ColIV (AbD Serotec 2150–1470, 1:100), rat anti-CD68 (Abcam ab53444, 1:200), goat anti-Olig2 (R&D Systems AF2418, 1:100), rabbit anti-MHC Class II (Abcam ab180779, 1:100), rat anti-CD45 (Becton Dickinson 550539, 1:200), and rat anti MBP (Abcam ab7349, 1:200). The following donkey-raised secondary antibodies (all in 1:400 dilution unless otherwise stated) were used for immunostaining: anti-rabbit IgG conjugated to Alexa Fluor (AF) 488 (Thermo Fisher Scientific A21206), anti-chicken IgY AF488 (Jackson ImmunoResearch 703-545-155), anti-goat IgG AF488 (Invitrogen, A-11055), anti-rat IgG Cy3 (Jackson ImmunoResearch 712-165-153), anti-rabbit IgG AF546 (Thermo Fisher Scientific A10040), anti-rat IgG AF594 (Thermo Fisher A21209), anti-rabbit IgG AF594 (Thermo Fisher Scientific A21207), anti-rabbit IgG AF647 (Thermo Fisher Scientific A31573), and anti-goat IgG AF647 (Thermo Fisher Scientific A21447). Streptavidin AF405 (Invitrogen S32351, 1:200) was used for detection of biotinylated-IB4 stained samples. Nuclei were counterstained with 4',6-diamidino-2-phenylindole (Sigma, D9542) diluted at 1 µg ml$^{-1}$ together with the secondary antibodies.

## Statistics and reproducibility

No statistical methods were used to predetermine sample size. The experiments were not randomized, and investigators were not blinded to allocate during experiments and outcome assessment.

Data sets with normal distributions were analysed with unpaired Student's two-tailed t-tests to compare two conditions. Results are depicted as mean ± s.e.m. as indicated in figure legends. All experiments for quantitative analysis and representative images were reproduced at least three times.

## Acknowledgements

This work was supported by the Max Planck Society, the University of Münster and the German Research Foundation (DFG) CRC1366A01, CRC1009A02, SO285/11-1, and the Cluster of Excellence 'Cells in Motion' (EXC1003).

## Additional information

### Funding

| Funder | Grant reference number | Author |
| --- | --- | --- |
| Max-Planck-Gesellschaft | | Ralf H Adams |
| University of Münster and the German Research Foundation | CRC1366A01 | Ralf H Adams |
| University of Münster and the German Research Foundation | CRC1009A02 | Ralf H Adams |
| University of Münster and the German Research Foundation | SO285/11-1 | Ralf H Adams |
| Cluster of Excellence 'Cells in Motion' | EXC1003 | Lydia Sorokin |

The funders had no role in study design, data collection and interpretation, or the decision to submit the work for publication.

### Author contributions

Hyun-Woo Jeong, Conceptualization, Formal analysis, Investigation, Methodology, Writing - original draft, Project administration, Writing - review and editing; Rodrigo Diéguez-Hurtado, Investigation, Methodology, Writing - review and editing; Hendrik Arf, Methodology; Jian Song, Formal analysis, Investigation, Methodology; Hongryeol Park, Investigation, Methodology; Kai Kruse, Formal analysis, Validation, Methodology; Lydia Sorokin, Supervision, Methodology, Writing - review and editing; Ralf H Adams, Conceptualization, Supervision, Funding acquisition, Writing - original draft, Project administration, Writing - review and editing

## Author ORCIDs
Hyun-Woo Jeong (ID) http://orcid.org/0000-0002-6976-6739
Rodrigo Diéguez-Hurtado (ID) http://orcid.org/0000-0002-2055-599X
Hendrik Arf (ID) http://orcid.org/0000-0002-2038-705X
Lydia Sorokin (ID) http://orcid.org/0000-0001-7704-7921
Ralf H Adams (ID) http://orcid.org/0000-0003-3031-7677

## Ethics
All animal experiments were performed in compliance with the relevant laws and institutional guidelines, approved by local animal ethics committees, and conducted with permissions (84-02.04.2016. A525, 81-02.04.2020.A471 and 84-02.04.2017.A322) granted by the Landesamt für Natur, Umwelt und Verbraucherschutz (LANUV) of North Rhine-Westphalia, Germany. Every effort was made to minimize suffering.

## Decision letter and Author response
Decision letter https://doi.org/10.7554/eLife.57520.sa1
Author response https://doi.org/10.7554/eLife.57520.sa2

## Data availability
Raw data (fastq files) and processed data (gene counts) for single cell RNA-Seq analysis have been deposited in the Gene Expression Omnibus with the primary accession number, GSE133283.

The following dataset was generated:

| Author(s) | Year | Dataset title | Dataset URL | Database and Identifier |
|---|---|---|---|---|
| Jeong H, Adams RH | 2019 | Single cell transcriptome analysis of non-neuronal cell populations in mouse brain cortex | https://www.ncbi.nlm.nih.gov/geo/query/acc.cgi?acc=GSE133283 | NCBI Gene Expression Omnibus, GSE133283 |

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
