## [Editor Report]

This study provides insight into an understudied cell type in the neurovascular unit involved in inflammatory disease and provides a resource of scRNA seq data for non-neuronal CNS cells. Further, it provides some interesting areas for future investigations into neuroimmune mechanisms used by the brain vasculature to control leukocyte transmigration in various health and disease conditions. This work will be of interest to vascular and neurovascular biologists, aging biologists, immunologists, and translational/clinical scientists interested in disease therapies.

---

## [Decision Letter]

**Decision letter after peer review:**

Thank you for submitting your article "Single cell transcriptomics reveals functionally specialized vascular endothelium in brain" for consideration by *eLife*. Your article has been reviewed by 2 peer reviewers, and the evaluation has been overseen by a Reviewing Editor and Carla Rothlin as the Senior Editor. The reviewers have opted to remain anonymous.

The reviewers have discussed the reviews with one another and the Reviewing Editor has drafted this decision to help you prepare a revised submission.

The editors have judged that your manuscript is of interest, but as described below extensive revision is required before it is published. We would like to draw your attention to changes in our revision policy that we have made in response to COVID-19 (https://elifesciences.org/articles/57162). First, because many researchers have temporarily lost access to the labs, we will give authors as much time as they need to submit revised manuscripts. We are also offering, if you choose, to post the manuscript to bioRxiv (if it is not already there) along with this decision letter and a formal designation that the manuscript is "in revision at *eLife*". Please let us know if you would like to pursue this option. (If your work is more suitable for medRxiv, you will need to post the preprint yourself, as the mechanisms for us to do so are still in development.)

Summary:

Jeong et al. employ single cell transcriptional profiling to address cellular heterogeneity of the neurovascular unit. By using the murine cerebral cortex as the anatomical site of interest, the authors dissect cellular heterogeneity within four distinct conditions: aging, inflammation, development and adulthood. This approach has revealed a potential new endothelial subset – reactive endothelial venules, which are post-capillary endothelial cells with specialized functions particularly for immune cell transmigration and potential communication with perivascular macrophages within the brain parenchyma.

This study provides insight into largely understudied cell types in the neurovascular unit. Further, it provides for some interesting areas of future investigations into neuroimmune mechanisms that are used by the brain vasculature to control leukocyte transmigration in various health and disease conditions. However, as presented the work is not well-described or annotated, some methodology is not clear, numerous functional claims that are not supported by the data, and relevant work of other labs is not acknowledged and incorporated into the interpretations. The suggestion is to reformat the work as a resource paper with appropriate attributions of work of others, remove extensive unsubstantiated claims, and significantly improve description of the methodology and description of the data.

Essential revisions:

1. What is the gap in knowledge that is filled by this work? The scope of the study is on PVECs and a more suitable introduction with relevant literature citation and gap of knowledge should be included. The current broad claim of "BBB heterogeneity" is not very relevant to the current experimental design and findings in this manuscript. The authors should introduce REVs early on as it is a main finding. Although this cell population is in early data (Figure 2A) the first description and discussion of this cell population is towards the end of the main text.

2. There are many areas of bold claims about probable functional roles without functional data to support these ideas. It is suggested to remove these claims and reformat as a resource paper, with appropriate attribution of other work in the field and comparison of analyses. Specifically: (a) the presented work is done using mice, while recently the human cell atlas was published (PMID: 32214235) that should be re-analyzed to find REVs and address relevance of the mouse model for studies of human disorders; (b) work would benefit from comparison of authors' dataset with previously published data, e.g. of Sabbagh MF et al., 2018 (PMID: 30188322), Vanlandewijck M et al., 2018 (PMID:29925939), He L et al., 2018 (PMID:30129931) or Kalucka J et al. 2020 (PMID: 32059779) to examine reproducibility of gene expression, transcriptomic signatures, and EC subtypes for relevant conditions; (c) recently Tony Wyss-Coray and colleagues reported that brain capillary, arterial, and venous cells age differently at the transcriptome level, that aged capillaries upregulate innate immunity and oxidative stress pathways, and that hippocampal neural precursor activity is impaired via endothelial VCAM1 (PMID: 32234477, PMID: 31086348). These results should be discussed and adequately compared to the newly obtained results. Also, there is speculation and claims about a functional module that may exist between ICAM1+ vessels and PVMs, with little molecular detail in terms of validation or functional data to support this hypothesis. Simply having cell types in the same region does not dictate that they communicate in meaningful ways, so this claim should be removed. The model suggests that perhaps LYVE1+ PVMs may actively suppress leukocytes that enter through REVs but there is nothing to support this, so this claim should be removed.

3. Readability is very low, with extensive lack of concordance between text and data, precluding rigorous assessment. The authors submitted 4 main figures and 9 extended data figures, but refer to 7 main figures and 10 (or 11 based on the figure legend numbering) extended data figures. Some of the figures are missing and panels in figures do not match text (and the other way round). Legends poorly describe the panels. Abbreviations in Figures are not explained. Some figures are overcrowded and presented data is not explained in the main text (e.g. for Figure 1d). Authors should provide rationale behind prioritising the data presented in the main figures. There are several mistakes in the figures themselves. For example, in Figure 1a panels are said to represent UMAP, but t-SNE plots are shown.

4. Some provided images do not support validation claims, e.g. Extended Data Figure 8 does not support statement that ICAM-1 expression in the brain parenchyma remains restricted to veins/venules and slightly expands into adjacent vessels. Double or triple staining to clearly illustrate veins/venules vs. arteries is needed to support these assumptions. Also, based on provided representative images in Extended Data Figure 7c, loss of MBP protein is more severe during the onset of EAE and not in its peak as authors describe in the text. How can the authors explain that? It is also unclear why the authors used Olig2 staining for these experiments.

5. Methodology is not clear. Seq analysis cell numbers are a concern – it is unclear how many cells were obtained in each subpopulations. Methods must include more technical detail. For example, in Figure 2, ~11,000 endothelial cells are analyzed from both adult and aged mice – it is unclear why these two groups are pooled. Tony Wyss-Coray recently reported via seq that brain ECs are transcriptionally different from aged mice, so this could argue that pooling adult and aged ECs into the same analysis is a potential confound. Further, it is unclear how many cells make up ICAM1+ and other subpopulations (I,e, what are the cell numbers of ICAM1+ PVECs and ICAM1-PVECs?). Limited cell counts from the brain vasculature is a known technical issue for sequencing, so cell numbers are important to understand whether methods are appropriate. Are there thousands, hundreds or dozens of ICAM1+ ECs from this overall population? Also, pooling of 3 PVM groups seems likely to lead to possible confounding factors. Why are these 3 groups chosen? The analysis actually indicates that these populations are likely unique (or at least somewhat different) across these conditions, so pooling may not be appropriate. Is there a biological response for this pooling? If cell numbers needed to be increased, would it be more appropriate to increase the number of mice per condition and keep them separate?

[Editors' note: further revisions were suggested prior to acceptance, as described below.]

Thank you for resubmitting your work entitled "Single cell transcriptomics reveals functionally specialized vascular endothelium in brain" for further consideration by *eLife*. Your revised article has been evaluated by Carla Rothlin (Senior Editor) and a Reviewing Editor.

The manuscript has been improved but there are some remaining issues that need to be addressed, as outlined below:

1. Addition of several control panels (Figure 2, Supp 3G; Figure 5E-F).

2. Further proof of T-cell accumulation into the brain under stimulated conditions (see Rev 2 comment below) via images and quantification, and/or careful use of words to exactly describe what is demonstrated by the data in several places.

3. Correction of typos.

For your convenience, the remarks from reviewers are appended below in their entirety.

*Reviewer #1 (Recommendations for the authors):*

The reviewer would like to thank the authors for considering the submitted remarks. The current version of the manuscript was an absolute pleasure to read. Although the manuscript is not perfect, for example, the authors could try to validate the results obtained from the ligand-receptor analysis or there are still unfinished sentences/typos, e.g. in lines #348 or #352, I do not have any major criticisms that will improve the paper if answered, only some that will delay publication.

I am confident that this study will lay the foundations for future work and will be an excellent resource for the broader scientific community.

*Reviewer #2 (Recommendations for the authors):*

In this revision, Jeong et al. have dramatically enhanced the rigor of this study and have addressed most of our major concerns originally brought forward. Notably, the overarching, unsubstantiated claims in the first version have been removed or have had significant data additions to help support these claims. Thank you for the addition of the discussion of how your scRNA-seq dataset compares to previously published datasets. Overall, the text is much more readable and the data progression is logical and clear. Prior to publication, there is still one major concern that needs to be addressed and we bring a few minor corrections below.

1. Figure 2, figure supplement 3G – the text in lines 161-162 claims that CD3+ T cells accumulate and penetrate the brain but the images do not clearly demonstrate this. Rather, it appears that they are accumulating, and perhaps not penetrating. This is a really important point to clarify – whether T cells are physically entering the brain at these sites needs to be explored. Better images showing the T cells in the brain parenchyma and/or quantification of the numbers of T cells accumulating and penetrating is warranted. It is also important to note there isn't a control here – there should be a negative (PBS) control to show that T cells in the LPS condition are actually different. EM would be able to prove these cells are transmigrating and/or flow cytometry quantification of T cell numbers in the brain parenchyma could be helpful.

Line 295 in the discussion states "recruitment", but again there is no direct evidence that REVs are actually allowing T cells to recruit to the brain. Even though these sites are likely to be hotspots of leukocyte accumulation, be cautious about using recruitment.

2. Similar to the comments above, for Figures 5 E and F, the authors should show the negative control side-by-side so the reader can really appreciate the differences.

3. In your response to reviewer comments: "LPS treatment promotes the adhesion of CD3+ cells to ICAM-1+ REVs (Figure 2-supplement 3G). While our findings indeed indicate that REVs are present in the absence of any challenge, leukocyte adhesion is extremely rare under baseline conditions. Thus, while we propose that REVs are poised for leukocyte transmigration, this is prevented by the BBB properties of the brain vasculature in the absence of inflammatory challenge."

Are you suggesting that LPS induces loss of BBB features and that controls the difference between the baseline activity of REVs and LPS-induced activity? If so, then presumably REVs are indeed able to allow for attachment of T cells in homeostatic conditions, but only under inflammatory conditions when the BBB is weakened, can recruitment actually occur? Some level of discussion on this topic could be helpful in the discussion.

---

## [Author Response]

Summary:Jeong et al. employ single cell transcriptional profiling to address cellular heterogeneity of the neurovascular unit. By using the murine cerebral cortex as the anatomical site of interest, the authors dissect cellular heterogeneity within four distinct conditions: aging, inflammation, development and adulthood. This approach has revealed a potential new endothelial subset – reactive endothelial venules, which are post-capillary endothelial cells with specialized functions particularly for immune cell transmigration and potential communication with perivascular macrophages within the brain parenchyma.This study provides insight into largely understudied cell types in the neurovascular unit. Further, it provides for some interesting areas of future investigations into neuroimmune mechanisms that are used by the brain vasculature to control leukocyte transmigration in various health and disease conditions. However, as presented the work is not well-described or annotated, some methodology is not clear, numerous functional claims that are not supported by the data, and relevant work of other labs is not acknowledged and incorporated into the interpretations. The suggestion is to reformat the work as a resource paper with appropriate attributions of work of others, remove extensive unsubstantiated claims, and significantly improve description of the methodology and description of the data.

We are grateful for this assessment of our work. As suggested by the reviewers, we have improved our manuscript and now present our work as a resource paper. As part of our revision, we have included additional references to published work and we have carefully reviewed all claims and conclusions.

Essential revisions:1. What is the gap in knowledge that is filled by this work? The scope of the study is on PVECs and a more suitable introduction with relevant literature citation and gap of knowledge should be included. The current broad claim of "BBB heterogeneity" is not very relevant to the current experimental design and findings in this manuscript.

We now provide a clear focus on reactive endothelial venules (REVs) as a specialized vessel subtype in the brain vasculature with relevance for neuroinflammation. In the revised manuscript, we have added a paragraph describing the relevance of postcapillary venules and the surrounding perivascular space in CNS immune responses to the Introduction. Furthermore, we now provide a better description of the properties of REVs (see also next question) and avoided “broad” claims about the BBB, which has not been a major subject of the current study.

The authors should introduce REVs early on as it is a main finding. Although this cell population is in early data (Figure 2A) the first description and discussion of this cell population is towards the end of the main text.

We now mention REVs at the end of the Introduction and, as suggested by the reviewer, describe this EC subpopulation in Figure 2.

2. There are many areas of bold claims about probable functional roles without functional data to support these ideas. It is suggested to remove these claims and reformat as a resource paper, with appropriate attribution of other work in the field and comparison of analyses. Specifically: (a) the presented work is done using mice, while recently the human cell atlas was published (PMID: 32214235) that should be re-analyzed to find REVs and address relevance of the mouse model for studies of human disorders;

We have added a paragraph to the Discussion in the revised manuscript and now discuss various previous studies in the context of our work. It has to be noted that many of the previous scRNA-seq studies for brain provide a rather limited coverage of ECs, reflecting the high abundance of neural cells, or have used positive selection (cell sorting) for the isolation of vascular cells. The latter will introduce a certain experimental bias, due to the expression of transgenic reporters, and will obviously miss cells lacking reporter expression.

The data generated by Han et al. (PMID: 32214235) and Human Cell Landscape database (http://bis.zju.edu.cn/HCL) is very useful, but contains only 74 vascular cells (ECs + pericytes) among the 8,531 adult brain cells (see Author response image 1). Therefore, sequential subclustering analysis identifying EC subsets including REV is not feasible with this data.

**Author response image 1. sa2fig1:** 

(b) work would benefit from comparison of authors' dataset with previously published data, e.g. of Sabbagh MF et al., 2018 (PMID: 30188322),

P7 brain EC heterogeneity/subclustering results reported by Sabbagh et al. (PMID: 30188322) are highly consistent with our P10 EC subclustering results (Figure 2—figure supplement 1). However, Sabbagh et al. did not address EC heterogeneity in adult or aged stages.

Vanlandewijck M et al., 2018 (PMID:29925939), He L et al., 2018 (PMID:30129931)

ECs in these studies were isolated by sorting based on GFP expression in Cldn5-GFP transgenic reporter mice, followed by sequencing using the SmartSeq2 method. However, these ECs are not clearly segregated into different subpopulations by unsupervised hierarchical clustering (e.g. K-nearest neighbor (KNN) graph-based clustering approach). Instead, authors ordered cells into one-dimensional range using the SPIN method based on known arteriovenous markers, which excludes the identification of minor subpopulations. Nevertheless, Icam1 and Vwf are enriched in a fraction of venous ECs (see Author response image 2), which is consistent with our own findings.

or Kalucka J et al. 2020 (PMID: 32059779) to examine reproducibility of gene expression, transcriptomic signatures, and EC subtypes for relevant conditions;

In addition to traditional EC phenotypes of venous to artery, Kalucka et al. (PMID: 32059779) identified an ‘interferon-activated EC population” in brain (6%), but this population is not consistent with REV. For example, Vwf is not detected in this population, which is, instead, enriched in clusters of large arteries and large veins.

(c) recently Tony Wyss-Coray and colleagues reported that brain capillary, arterial, and venous cells age differently at the transcriptome level, that aged capillaries upregulate innate immunity and oxidative stress pathways, and that hippocampal neural precursor activity is impaired via endothelial VCAM1 (PMID: 32234477, PMID: 31086348). These results should be discussed and adequately compared to the newly obtained results.

Agree. We now discuss these previously published data in the context of our own work. While this was not a main focus of our study, we confirm the expression of pro-inflammatory genes in aged ECs similar to what has been reported by Chen et al. (PMID: 32234477).

Also, there is speculation and claims about a functional module that may exist between ICAM1+ vessels and PVMs, with little molecular detail in terms of validation or functional data to support this hypothesis. Simply having cell types in the same region does not dictate that they communicate in meaningful ways, so this claim should be removed. The model suggests that perhaps LYVE1+ PVMs may actively suppress leukocytes that enter through REVs but there is nothing to support this, so this claim should be removed.

In the revised manuscript, we present our findings such as the association of PVMs with ICAM1+ vessels and the differences between PVM subpopulations without speculating about potential functional consequences. We state, for example, that “Lyve1+ PVMs express anti-inflammatory or immune-suppressive (M2 macrophage-like) polarization genes” without claiming that these cells may actively suppress leukocytes.

3. Readability is very low, with extensive lack of concordance between text and data, precluding rigorous assessment. The authors submitted 4 main figures and 9 extended data figures, but refer to 7 main figures and 10 (or 11 based on the figure legend numbering) extended data figures. Some of the figures are missing and panels in figures do not match text (and the other way round). Legends poorly describe the panels. Abbreviations in Figures are not explained. Some figures are overcrowded and presented data is not explained in the main text (e.g. for Figure 1d). Authors should provide rationale behind prioritising the data presented in the main figures. There are several mistakes in the figures themselves. For example, in Figure 1a panels are said to represent UMAP, but t-SNE plots are shown.

We apologize for any inconsistencies in the original submission. As part of our revision, we have carefully checked the manuscript to ensure that text, figures and legends match each other.

4. Some provided images do not support validation claims, e.g. Extended Data Figure 8 does not support statement that ICAM-1 expression in the brain parenchyma remains restricted to veins/venules and slightly expands into adjacent vessels. Double or triple staining to clearly illustrate veins/venules vs. arteries is needed to support these assumptions.

We have added immunostaining data showing αSMA, ICAM-1 and CD31 (Figure 5C), which confirms venous-restricted ICAM-1 expression.

Also, based on provided representative images in Extended Data Figure 7c, loss of MBP protein is more severe during the onset of EAE and not in its peak as authors describe in the text. How can the authors explain that? It is also unclear why the authors used Olig2 staining for these experiments.

Disease onset or peak of disease are determined based on symptoms displayed by the mice in the experiment. These symptoms are preceded by pathogenic changes in cellular or molecular level (e.g. loss of myelin (MBP) or apoptosis of oligodendrocytes). EAE is a transient model, which, depending on the severity of symptoms, either results in the death or the full or partial recovery of the affected animals (remission) (PMID: 18432984; PMID: 24362236; PMID: 21371012).

Immunization against myelin oligodendrocyte glycoprotein (MOG) is typically used for the induction of EAE and oligodendrocyte death is a feature of both multiple sclerosis and EAE.

Olig2 was used as a marker of oligodendrocytes.

5. Methodology is not clear. Seq analysis cell numbers are a concern – it is unclear how many cells were obtained in each subpopulations. Methods must include more technical detail. For example, in Figure 2, ~11,000 endothelial cells are analyzed from both adult and aged mice – it is unclear why these two groups are pooled. Tony Wyss-Coray recently reported via seq that brain ECs are transcriptionally different from aged mice, so this could argue that pooling adult and aged ECs into the same analysis is a potential confound. Further, it is unclear how many cells make up ICAM1+ and other subpopulations (I,e, what are the cell numbers of ICAM1+ PVECs and ICAM1-PVECs?). Limited cell counts from the brain vasculature is a known technical issue for sequencing, so cell numbers are important to understand whether methods are appropriate. Are there thousands, hundreds or dozens of ICAM1+ ECs from this overall population?

We have added information about cell numbers and subcluster proportions in each plot. Integration of different datasets was necessary for the identification of conserved populations and to investigate further detailed analysis including the DEG test at subcluster resolution.

It is unclear how many postcapillary venules are ICAM1+ REVs, because scRNA-seq does not necessarily provide an accurate readout about cell numbers. Certain cells might more easily recovered than others during the procedure and, conversely, certain cells might be more easily damaged or lost.

Nevertheless, we assume that not all postcapillary venules are REVs because REV ECs with clear molecular characteristics represent only 3% (~400 cells) in our scRNA-seq data. ICAM1-negative non-REV postcapillary venules are not distinguishable from CapV-Venous EC continuum and may not represent a distinct subpopulation.

Also, pooling of 3 PVM groups seems likely to lead to possible confounding factors. Why are these 3 groups chosen? The analysis actually indicates that these populations are likely unique (or at least somewhat different) across these conditions, so pooling may not be appropriate. Is there a biological response for this pooling? If cell numbers needed to be increased, would it be more appropriate to increase the number of mice per condition and keep them separate?

Cell isolation, library preparation and sequencing were done independently for each sample set and the resulting data sets were subsequently integrated to facilitate the identification of differences and common features. Data integration is very informative for subcluster identification and direct comparison of cells from different age or treatment groups.

[Editors' note: further revisions were suggested prior to acceptance, as described below.]

Thank you for resubmitting your work entitled "Single cell transcriptomics reveals functionally specialized vascular endothelium in brain" for further consideration by eLife. Your revised article has been evaluated by Carla Rothlin (Senior Editor) and a Reviewing Editor.The manuscript has been improved but there are some remaining issues that need to be addressed, as outlined below:1. Addition of several control panels (Figure 2, Supp 3G; Figure 5E-F).

Following your feedback, we have added negative control panels to Figure2—figure supplement 3G as well as in Figure 5E. Control panels have been also added to Figure 5F, which has been repositioned to Figure 5—figure supplement 3B.

2. Further proof of T-cell accumulation into the brain under stimulated conditions (see Rev 2 comment below) via images and quantification, and/or careful use of words to exactly describe what is demonstrated by the data in several places.

As suggested, we have revised our description of events relating to leukocyte trafficking. In particular, we replaced the term “recruitment” with “accumulation” in the discussion panel. In our description of Figure 2—figure supplement 3G, we replaced the term “leukocyte infiltration” with “leukocyte trafficking” and removed the word “penetration”.

In addition, we performed flow cytometry to quantify the brain-infiltrated leukocytes in the EAE model. After removal of circulating cells by robust perfusion, we collected mononuclear cells using a Percoll gradient from the crushed brain tissues without enzymatic dissociation. This showed a striking increase in CD45^+^ leukocytes at EAE onset and peak. The new data is shown in Figure 5—figure supplement 3C.

Reviewer #1 (Recommendations for the authors):The reviewer would like to thank the authors for considering the submitted remarks. The current version of the manuscript was an absolute pleasure to read. Although the manuscript is not perfect, for example, the authors could try to validate the results obtained from the ligand-receptor analysis or there are still unfinished sentences/typos, e.g. in lines #348 or #352, I do not have any major criticisms that will improve the paper if answered, only some that will delay publication.I am confident that this study will lay the foundations for future work and will be an excellent resource for the broader scientific community.

We are grateful to the reviewer for the constructive input and positive evaluation. Of course, our study is not perfect but it is an intrinsic feature of scientific publications that there always limitations and open questions.

Reviewer #2 (Recommendations for the authors):In this revision, Jeong et al. have dramatically enhanced the rigor of this study and have addressed most of our major concerns originally brought forward. Notably, the overarching, unsubstantiated claims in the first version have been removed or have had significant data additions to help support these claims. Thank you for the addition of the discussion of how your scRNA-seq dataset compares to previously published datasets. Overall, the text is much more readable and the data progression is logical and clear. Prior to publication, there is still one major concern that needs to be addressed and we bring a few minor corrections below.

We are very grateful for this assessment and have used the feedback to improve the manuscript further.

1. Figure 2, figure supplement 3G – the text in lines 161-162 claims that CD3+ T cells accumulate and penetrate the brain but the images do not clearly demonstrate this. Rather, it appears that they are accumulating, and perhaps not penetrating. This is a really important point to clarify – whether T cells are physically entering the brain at these sites needs to be explored. Better images showing the T cells in the brain parenchyma and/or quantification of the numbers of T cells accumulating and penetrating is warranted. It is also important to note there isn't a control here – there should be a negative (PBS) control to show that T cells in the LPS condition are actually different. EM would be able to prove these cells are transmigrating and/or flow cytometry quantification of T cell numbers in the brain parenchyma could be helpful.Line 295 in the discussion states "recruitment", but again there is no direct evidence that REVs are actually allowing T cells to recruit to the brain. Even though these sites are likely to be hotspots of leukocyte accumulation, be cautious about using recruitment.

As suggested, we have revised our description of events relating to leukocyte trafficking. In particular, we replaced the term “recruitment” with “accumulation” in the discussion panel. In our description of Figure 2—figure supplement 3G, we replaced the term “leukocyte infiltration” with “leukocyte trafficking” and removed the word “penetration”.

In addition, we performed flow cytometry to quantify the brain-infiltrated leukocytes in the EAE model. After removal of circulating cells by robust perfusion, we collected mononuclear cells using a Percoll gradient from the crushed brain tissues without enzymatic dissociation. This showed a striking increase in CD45^+^ leukocytes at EAE onset and peak. The new data is shown in Figure 5—figure supplement 3C.

2. Similar to the comments above, for Figures 5 E and F, the authors should show the negative control side-by-side so the reader can really appreciate the differences.

As suggested, we have added negative control panels to Figure 2—figure supplement 3G as well as in Figure 5E. Control panels have been also added to Figure 5F, which has been repositioned to Figure 5—figure supplement 3B.

3. In your response to reviewer comments: "LPS treatment promotes the adhesion of CD3+ cells to ICAM-1+ REVs (Figure 2-supplement 3G). While our findings indeed indicate that REVs are present in the absence of any challenge, leukocyte adhesion is extremely rare under baseline conditions. Thus, while we propose that REVs are poised for leukocyte transmigration, this is prevented by the BBB properties of the brain vasculature in the absence of inflammatory challenge."Are you suggesting that LPS induces loss of BBB features and that controls the difference between the baseline activity of REVs and LPS-induced activity? If so, then presumably REVs are indeed able to allow for attachment of T cells in homeostatic conditions, but only under inflammatory conditions when the BBB is weakened, can recruitment actually occur? Some level of discussion on this topic could be helpful in the discussion.

We propose that REVs facilitate the transmigration of activated leukocyte during neuroinflammation and are already primed for this role under homeostatic conditions without immune cell activation. In the healthy (unchallenged) brain, the adhesion and accumulation of leukocytes in REVs are extremely rare events because immune cells are not activated and LFA-1, a receptor for ICAM-1 expressed on leukocytes, has a low affinity for ICAM-1. However, reports in the literature support that “innate and adaptive immune cells are…known to have protective/healing properties in the CNS, as long as their activity is regulated, and their recruitment is well controlled” (quote taken from a review by Schwartz et al., PMID: 24198349). It therefore seems possible that REVs are able to mediate the attachment of T cells to the brain vasculature in homeostatic conditions and might even control the rare entry of these cells into the brain parenchyma. Immune cell accumulation is greatly enhanced under inflammatory conditions when the BBB is weakened. The expansion of REVs might contribute to this accumulation together with other factors such as the release of chemokines and the activation of immune cells.

We have added a statement to the Discussion on page 12 of the manuscript.